# Evidence that endosperm turgor pressure both promotes and restricts seed growth and size

Audrey Creff[1,2], Olivier Ali [1,2] ✉, Camille Bied[1], Vincent Bayle [1], Gwyneth Ingram [1] ✉ & Benoit Landrein [1] ✉

In plants, as in animals, organ growth depends on mechanical interactions between cells and tissues, and is controlled by both biochemical and mechanical cues. Here, we investigate the control of seed size, a key agronomic trait, by mechanical interactions between two compartments: the endosperm and the testa. By combining experiments with computational modelling, we present evidence that endosperm pressure plays two antagonistic roles: directly driving seed growth, but also indirectly inhibiting it through tension it generates in the surrounding testa, which promotes wall stiffening. We show that our model can recapitulate wild type growth patterns, and is consistent with the small seed phenotype of the *haiku2* mutant, and the results of osmotic treatments. Our work suggests that a developmental regulation of endosperm pressure is required to prevent a precocious reduction of seed growth rate induced by force-dependent seed coat stiffening.

How tissue growth arrest is achieved once an organ has reached a defined size is a key, yet unresolved, question in developmental biology[1,2]. In *Drosophila*, mechanical and biochemical signals have been proposed to act in concert to control growth and determine organ size in the wing imaginal disk[3,4]. In plants, mechanical signals can affect growth by modulating key processes such as cytoskeleton organization[5,6], auxin distribution[7,8], chromatin organization[9] and gene expression[10,11]. However, it remains unclear whether mechanical signals are involved in organ size control in plants.

Seed size is a key agronomic trait that influences seed composition, and viability[12]. Seed growth relies on interactions between two seed compartments: the endosperm and the testa[13] (Fig. 1a). During early post-fertilization development in *Arabidopsis*, the endosperm comprises a single poly-nucleate cell filling most of the internal compartment of the seed[14]. Hydrostatic pressure (turgor) in the endosperm, resulting from osmolite accumulation, is thought to drive seed growth, while progressive reduction of endosperm turgor was proposed to contribute to seed growth arrest[15].

The testa, a maternal tissue derived from the ovule chalaza and integuments[16], is thought to constrain seed growth. During mid to late seed expansion, the adaxial epidermis of the outer-integument (ad-oi) of the testa appears to restrict growth by reinforcing its inward-facing cell wall (wall 3, the third periclinal wall counting from the outside, Supplementary Fig. 1a). This process could involve the perception of tensile stresses induced in the testa by endosperm pressure. Indeed, the expression of *ELA1 (EUI-LIKE P450 A1)*, a negative seed size regulator expressed predominantly in ad-oi[17], is promoted by increasing tensile stress in this layer[11].

Here, we test whether seed size could be determined by a mechanosensitive incoherent feedforward loop in which the direct growth-promoting activity of endosperm turgor is antagonized by an indirect growth inhibition resulting from the mechanosensitive stiffening of testa walls. We show that our model can be used to explain the small seed phenotype of a mutant with abnormal endosperm pressure, *haiku2 (iku2)*, as well as the effects of manipulations of turgor pressure in seeds growing in vitro. Finally, we show that the small seed size phenotype of the *iku2* mutant can be partly complemented by altering testa wall differentiation in a mutant with defective outer-integument identity, *apetala2 (ap2)*. Our work sheds new light on the contribution of turgor pressure to plant organ growth.

[1]Laboratoire Reproduction et Développement des Plantes, Université de Lyon, ENS de Lyon, UCB Lyon, CNRS, INRAE, INRIA, F-69342 Lyon69364 Cedex 07France. [2]These authors contributed equally: Audrey Creff, Olivier Ali. ✉e-mail: olivier.ali@inria.fr; gwyneth.ingram@ens-lyon.fr; benoit.landrein@ens-lyon.fr

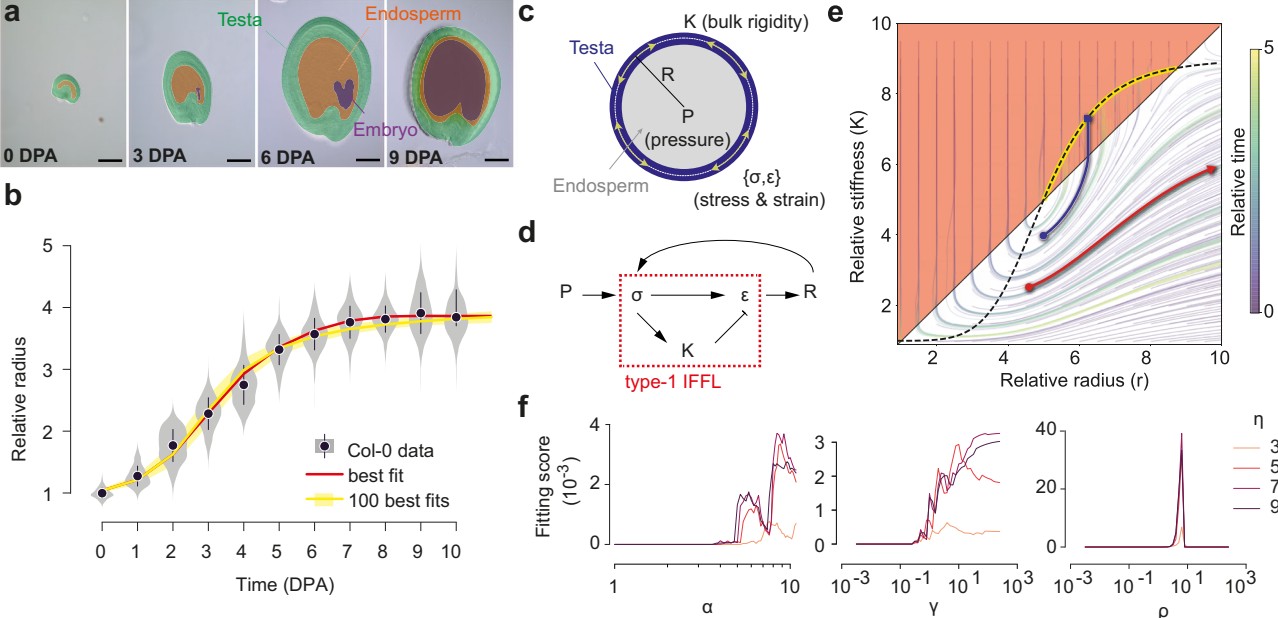

**Fig. 1 | Modeling seed growth using a mechanosensitive incoherent feedforward loop. a** Representative WT seeds (Col-0 ecotype) at different days postanthesis (DPA). The seed compartments are highlighted using false colors. Scale bars = 100 μm. **b** Relative seed radius as a function of time (DPA). Gray violin plots and black points correspond to the biological data (Col-0, pool of 6 independent experiments, 0DPA: *n* = 428, 1DPA: *n* = 485, 2DPA: *n* = 529, 3 DPA *n* = 483, 4 DPA: *n* = 510, 5 DPA: *n* = 511, 6 DPA: *n* = 505; 7 DPA: *n* = 493, 8 DPA: *n* = 501, 9 DPA: *n* = 446, 10 DPA: *n* = 447). The yellow line and yellow band depict the dynamics of the 100 simulations that fit best the experimental data, while the red line represents the simulation that best fits the data. **c**, **d** Modeling seed growth with an incoherent feedforward loop (IFFL). **e** Example of a representative stream map showing the evolution of the system over time within the relative radius/rigidity state space. All trajectories start from different sets of initial values of rigidity $K_O$ and radius $r_O$ and

evolve according to the same set of parameter values (α = 1, γ = 5, η = 5, ρ = 8). The area in pink corresponds to the zone of no growth, while the black dotted sigmoid curve marks the arrest of stiffening. The trajectories hitting the yellow part of this sigmoid curve (such as the one depicted in dark blue) correspond to simulations converging toward a steady state for both growth and stiffening. The trajectories oriented toward the right side of the graph (such as that depicted in red) correspond to diverging simulations that will not reach any steady state. **f** Fitting score of the simulations to the biological data (Col-0) as a function of four parameters characterizing stress-dependent shell stiffening: (α) amplitude of stiffening, (ρ) ratio between stiffening and growth thresholds, (η) steepness of the stiffening mechanism (Hill function exponent), (γ) characteristic time ratio between growth and stiffening.

## Results

### Quantitative analysis of WT seed growth pattern

We first analyzed wild-type (WT, ecotype Col-0) seed growth at 24 h intervals from anthesis using Differential Interference Contrast (DIC) imaging and seed size measurements (Fig. 1a). We observed that seed radius increases about 3.5 times, following a typical and reproducible S-curve, to plateau at around 7 days post-anthesis (DPA, Fig. 1b and Supplementary Fig. 1b). By deriving seed size measurements, we observed that seed growth peaked at 1–3 DPA before slowly decreasing between 3 and 7 DPA (Supplementary Fig. 1c). At the end of the growth phase, the endosperm undergoes cellularization, a progressive process which is necessary for subsequent embryo development[18]. Cellularization has been proposed to influence seed growth arrest because its onset correlates with the end of the growth phase in WT and in some seed size mutants[13,19]. However, we found that seed growth arrest is gradual and starts at least 2 days before the onset of cellularization, which occurs around 5 DPA (Supplementary Figs. 1c and 2a). We also analyzed seed growth in *ede1-3* (*endosperm defective 1*), a mutant lacking a microtubule-binding protein required for endosperm cellularization[18,20]. The *ede1-3* mutants showed defects in embryo development but only a minor seed growth defect (Supplementary Fig. 2b–d), supporting the idea that endosperm cellularization and seed growth arrest can be dissociated.

### Developing a mechanical model of seed growth

We next tested whether a model where endosperm pressure directly drives seed growth but indirectly inhibits it through stress-dependent

testa stiffening could explain WT seed growth patterns. To this end, we approximated the testa to a linearly elastic spherical shell of radius *R*, constant thickness *h* and homogeneous effective rigidity *K* (Fig. 1c). We considered two mechanosensitive mechanisms taking place within this system (i.e., the idealized testa): (i) stress-based wall stiffening and (ii) strain-based wall expansion. Taken together, both mechanisms can be formalized as a set of two dimensionless coupled differential equations, Eq. (1); where the first line depicts the strain-based growth process while the second line depicts the stress-based stiffening process:

$$\begin{cases} \dot{r} = \left(\frac{pr}{k} - 1\right)_+ r \\ \dot{k} = \gamma\left(1 - k + \alpha h_\eta(pr, \rho)\right) \end{cases} \quad (1)$$

In Eq. (1), $\left(\frac{pr}{k} - 1\right)_+$ and $h_\eta(pr, \rho)$ correspond to the two mechanosensitive functions considered. The former describes a threshold linear function while the latter corresponds to a classic Hill function, as commonly used in biochemical signaling pathways. The state of our system is described by a set of two dimensionless variables: $r = R/h$, a dimensionless measure of the radius of our system and *k* a dimensionless version of its effective rigidity *K*. Besides these two variables, Eq. (1) features four dimensionless parameters {γ, α, ρ, η} that quantify cell wall stiffening properties (speed, strength, threshold and steepness) relative to the growth process (Supplementary Note 1.4). Finally, the adimensional pressure *P* plays the role of an external control variable, as it is not a property of the system itself (i.e., the testa) but an input in the system (Supplementary Note 1).

In this model, the stress and strain borne by the testa are respectively proportional to $\rho r$ and $\rho r/k$. The fact that the strain is inversely proportional to the effective stiffness $k$ is the cornerstone of our model: an increase in stiffness will decrease strain and consequently curb growth. By promoting stiffening, pressure induced stresses therefore have antagonistic effects on testa growth. We formalized this system, schematized in Fig. 1d, as an incoherent mechanosensitive feedforward mechanism because its motif is similar to that displayed by a type-1 incoherent feedforward loop (IFFL) in gene regulatory networks, and because WT seed growth pattern also resembles the pulse-like output produced by type1-IFFL[21] (Supplementary Fig. 1c).

We first looked at the kind of dynamics that Eq. (1) could generate. To this end, we performed a steady state analysis (Supplementary Note 2 and Supplementary Fig. 3) that revealed two key properties: (i) steady states result from a balance between growth and stiffening. In the first line of Eq. (1), stress ($\rho r$) promotes growth, but as it is proportional to the system radius ($r$), this first line alone results in an unrealistic exponential increase of the system radius. The second line of Eq. (1) describes the increase of the stiffening rate of the testa induced by stress. If the effective stiffness ($k$) increases fast enough, it can balance the growth-induced increase in stress in the first line of Eq. (1) and cause strain to drop below the growth threshold, ultimately causing the system to reach a steady state. The existence of steady states is only dependent on the values of three parameters $\{\alpha, \rho, \eta\}$ (Supplementary Note 2 and Supplementary Fig. 3b–g). However, for a given set of parameter values allowing the existence of steady states, not all initial states can converge towards these steady states. Some follow unrealistic diverging trajectories in the state space (Fig. 1e). (ii) The endosperm pressure value has no influence on the ability of the system to reach a steady state (Supplementary Fig. 3i), confirming that endosperm pressure mainly acts as an external control variable determining final seed size (Supplementary Note 2).

We next performed a detailed parameter space exploration to better situate boundaries within which simulations yield results compatible with experimental measurements (Fig. 1e, f). Among the $5 \times 10^5$ parameter sets we tested, less than $2 \times 10^3$ yielded simulations converging toward biologically relevant steady-state solutions (i.e., simulations reaching a steady-state with a final radius $R_{10DPA}/R_{0DPA} = 3.5 \pm 0.5$). We quantitatively compared each of these $2 \times 10^3$ simulations with experimental data and scored their fit, keeping only the 100 best-fitting simulations (Fig. 1b). Taken together, the steady-state analysis and the analysis of the relationship between fitting score and parameter distribution revealed that the stress-sensitive stiffening would have to be highly non-linear (i.e., strong and sharp) to recapitulate experimental results. More importantly, the parameter space analysis also revealed that the ratio between growth and stiffening thresholds ($\rho$) was the most constrained parameter. The sharp peak of ($\rho$) at the value of 1 (Fig. 1f) notably suggests that stiffening needs to be late compared to growth, but also that a tight synchronization between both processes is required to allow seed growth control through a mechanosensitive regulation. Fitting our simulations to *ede1-3* seed growth dynamics retrieved similar parameter value distributions to those of the WT fit, underlining the similarity between WT and *ede1-3* seed growth patterns (Supplementary Fig. 4a, b).

### Evidence that higher endosperm pressure leads to reduced seed growth in the *iku2* mutant

We next investigated whether we could use our model to explain the phenotype of known seed size mutants. We analyzed a mutant allele of *HAIKU2 (IKU2)*, which encodes a receptor-like kinase only expressed in the endosperm (Fig. 2a), and acting in a zygotic growth-control pathway[22]. We observed that *iku2* seeds initially grew similarly to WT seeds but that after a few days, growth decreased faster in *iku2* than it

did in the WT, ultimately leading to the production of smaller seeds (Fig. 2b, c and Supplementary Fig. 5a). As *IKU2* is expressed in the endosperm, we hypothesized that *iku2* seed growth defects might result from altered endosperm turgor[15]. Using a published method to extract endosperm turgor from force-displacement curves obtained by nanoindentation[15], we confirmed that endosperm pressure decreases throughout the growth phase in WT seeds (Fig. 2d and Supplementary Fig. 5b–d). To our surprise, this decrease was less pronounced in *iku2* mutants so that *iku2* pressure was generally more elevated than that in the WT during most of the growth phase (from around 3 to 7 DPA).

We next tested if the shallower drop in endosperm pressure observed in *iku2* could explain its growth phenotype in simulations. First, we fitted the model at constant pressure to the experimental measurements of *iku2* growth, which did not alter the global behavior of the system (Supplementary Figs. 4a, c and 5e, f and Supplementary Table 5). We then applied a drop function to reproduce qualitatively the pressure reduction observed experimentally in WT seeds in the *iku2*-fitted simulations (Supplementary Note 3 and Supplementary Fig. 5e). Strikingly, we observed that reducing pressure led to an extension of the growth phase in the *iku2*-fitted simulations, which could now recapitulate experimentally measured WT growth patterns (Supplementary Fig. 5e, f). To go further, we then extracted time-dependent functions to quantitatively fit the average pressure drop of each genotype from all our experimental replicates pooled together; we also complemented these two data-fitted functions with three intermediate ones (Fig. 2e and Supplementary Note 3). We ran a parameter space exploration using the *iku2* pressure function as a reference input and extracted the simulations that best fitted the experimental measurements of *iku2* growth. Using the parameters from these simulations, we then used the WT and the intermediate pressure drops as inputs. We observed that increasing the intensity of the pressure drop in *iku2*-fitted simulations enhanced growth (Fig. 2f). Seed growth was more sensitive to the strength of the drop of pressure in simulations than observed experimentally, which may result from the fact that the model does not take into account the complexity of testa structure so that stress is directly proportional to pressure in the simulations.

Given that endosperm pressure directly promotes growth but indirectly inhibits it through force-dependent testa stiffening in our model, we hypothesized that the counterintuitive effect of pressure that we observed on seed growth, both in simulations and experimentally in *iku2*, could be linked to changes in testa mechanical properties. We first looked at the effect of pressure on testa stiffness in our simulations. We observed that increasing the depth of the drop of endosperm pressure to mimic that observed in the WT delayed the stiffening of the testa in *iku2*-fitted simulations (Fig. 3a and Supplementary Fig. 6a). We thus tested experimentally whether *iku2* seed growth defects could be due to precocious stress-dependent testa stiffening. We have previously shown, and here confirmed again (Supplementary Fig. 6d), that the expression of the mechanosensitive gene *ELA1* is increased in the *iku2* mutant by qPCR[11]. To be certain that *ELA1* is indeed over-expressed in *iku2* oi-ad layer, we quantified *pELA1::3X-VENUS-N7* reporter fluorescence in WT and *iku2* seeds. We observed higher fluorescence in the oi-ad layer of *iku2* mutant seeds than in WT seeds at all relevant stages of development, suggesting that increased endosperm pressure leads to an enhanced mechanical response of the testa in *iku2* (Fig. 3b, c and Supplementary Fig. 6b,c). We then addressed possible alterations of the mechanical properties of testa walls in *iku2* seeds.

The pectin matrix is thought to be a key determinant of cell wall mechanical properties. Homogalacturonans (HG), the most abundant pectins, are deposited in a methyl-esterified state, but can subsequently be demethylesterified through the action of pectin methylesterases (PMEs)[23]. In some cases, this process can promote enzymatic

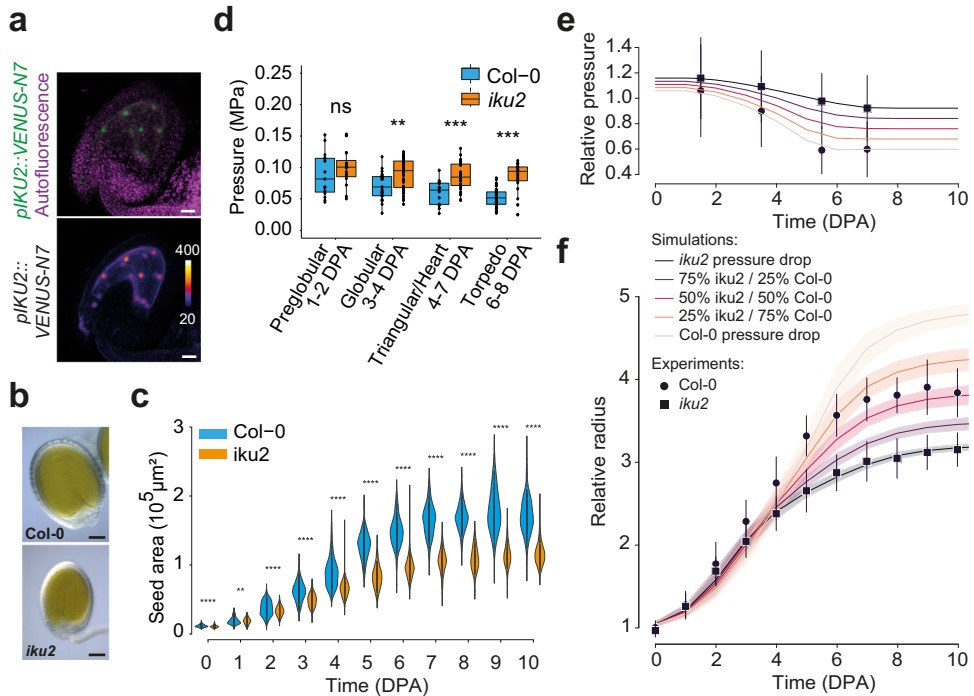

**Fig. 2 | Quantification of endosperm pressure and of seed growth pattern in
*iku2*. a** Expression of *pIKU2::VENUS-N7* in the endosperm of seeds at 1 day post-
anthesis (z-sum projection of a confocal stack), scale bar: 10 µm (*n* = 15 seeds from 2
insertions). **b** Representative Col-0 and *iku2* seeds at 10 DPA. Scale bars: 100 µm.
**c** WT and *iku2* seed growth patterns (Col-0, 6 independent experiments: 0DPA:
*n* = 428, 1DPA: *n* = 485, 2DPA: *n* = 529, 3 DPA: *n* = 483, 4 DPA: *n* = 510, 5 DPA: *n* = 511, 6
DPA: *n* = 505, 7 DPA: *n* = 493, 8 DPA *n* = 501, 9 DPA: *n* = 446, 10 DPA: *n* = 447; *iku2*, 4
independent experiments: 0DPA: *n* = 275, 1DPA: *n* = 280, 2DPA: *n* = 355, 3 DPA:
*n* = 324, 4 DPA: *n* = 360, 5 DPA: *n* = 369, 6 DPA: *n* = 353, 7 DPA: *n* = 322, 8 DPA *n* = 304,
9 DPA: *n* = 237, 10 DPA: *n* = 336). The black bars show the SD. Areas were compared
using two-sided Student tests without adjustments for multiple comparisons,
**p < 0.01, **p < 0.001, ****p < 0.0001. **d** Extraction of Col-0 and *iku2* endosperm
pressure through stiffness measurements performed by nanoindentation (Col-0:
preglobular: *n* = 15, globular: *n* = 23, triangular/heart: *n* = 14, torpedo: *n* = 51; *iku2*:
preglobular: *n* = 16, globular: *n* = 51, triangular/heart: *n* = 37, torpedo: *n* = 22). The
centerline shows the median; the box limits show the upper and lower quartiles, the
whiskers correspond to 1.5× interquartile range. Individual points are super-
imposed in black. Pressure values were compared using two-sided Student tests
without adjustments for multiple comparisons, **p < 0.01, ***p < 0.001. **e** Time-
dependent pressure functions used as inputs in simulations. Two pressure func-
tions were directly fitted from experimental measurements (pool of 4 independent
experiments, labeled *iku2* and Col-0 pressure drops, Col-0: preglobular: *n* = 31,
globular: *n* = 109, triangular/heart: *n* = 84, torpedo: *n* = 100; *iku2*: preglobular:
*n* = 51, globular: *n* = 89, triangular/heart: *n* = 76, torpedo: *n* = 106); three others were
interpolated as linear combinations of these two functions (resp. 75–25%, 50–50%
and 25–75%). **f** Relative seed radius as a function of time using the pressure drop
functions displayed in e as input. All simulations were performed over the 100-
parameter value sets best-fitting *iku2* experimental data extracted using the *iku2*
pressure drop function presented in e as an input (Col-0, 6 independent experi-
ments: 0DPA: *n* = 428, 1DPA: *n* = 485, 2DPA: *n* = 529, 3 DPA: *n* = 483, 4 DPA: *n* = 510, 5
DPA: *n* = 511, 6 DPA: *n* = 505, 7 DPA: *n* = 493, 8 DPA *n* = 501, 9 DPA: *n* = 446, 10 DPA:
*n* = 447; *iku2*, 4 independent experiments: 0DPA: *n* = 275, 1DPA: *n* = 280, 2DPA:
*n* = 355, 3 DPA: *n* = 324, 4 DPA: *n* = 360, 5 DPA: *n* = 369, 6 DPA: *n* = 353, 7 DPA: *n* = 322,
8 DPA *n* = 304, 9 DPA: *n* = 237, 10 DPA: *n* = 336). Plain curves display the mean
dynamics and confidence intervals show standard deviations. In both panels,
markers depict averaged experimental measurements and error bars standard
deviations.

HG degradation[23], weakening the cell wall and promoting growth[24].
However, fully demethylesterified HGs often form calcium-dependent
cross-links that increase wall stiffness and inhibit growth[25]. Using three
different antibodies (LM19, JIM5 and 2F4), we assessed HG methyles-
terification in seeds at different growth stages by immunolocalization
and subsequent signal quantification in the outer periclinal walls of the
testa using a custom-made pipeline (Supplementary Fig. 7). JIM5 pre-
ferentially detects pectins with low levels of methylesterification[26]
while LM19 and 2F4 preferentially detect demethylesterified
pectins[27,28]. In WT seeds, epitopes for all three antibodies were more
abundant in wall 3 than in other walls, supporting its load-bearing
role[11] (Fig. 3d–i and Supplementary Figs. 8–10). At 3 and 4 DPA, all
signals were weak and spotty (especially for JIM5 and 2F4), but
strongly increased between 4 and 6 DPA, consistent with model
predictions that testa wall stiffening occurs late compared to growth,
and should be strong and sharp (Fig. 3a). Strikingly, we also observed
that the signal was stronger in *iku2* than in Col-0 at early stages of
development (from 3 to 5 DPA depending on the antibody) but that it
was similar at the end of the growth phase for JIM5 and 2F4 (between
7 and 9 DPA).

As wall 3 is an internal wall embedded within the testa, it is not
possible to use atomic force microscopy to quantify the rigidity of the
wall in vivo. We thus tested if the changes in wall composition we
observed in *iku2* correlated with changes in wall resistance to rupture
by indentation. By adapting a previously published protocol[29], we
performed indentations of various depth on seeds at 5 DPA using a
conical tip, and quantified the frequency of testa wall failure using the
*LTi6b-GFP* membrane marker and Propidium Iodide (PI) staining
(Fig. 3j and Supplementary Fig. 11a). For all indentation depths, more
force was needed to indent *iku2* seeds than WT seeds (Supplementary
Fig. 11b), which likely results from the increased endosperm pressure
we measured in *iku2* at 5-6 DPA (Fig. 2d). We also observed that the
frequency of testa wall failure correlated with the indentation depth,
and that walls 1 and 2 were more easily ruptured than wall 3 (Fig. 3k and
Supplementary Fig. 11c). Finally, with an indentation of 40 µm, we
observed that the wall 3 of *iku2* seeds was significantly less prone to
rupture than that of the WT (*p* value <0.00001 in a $\chi^2$ test, Fig. 3k),
even though *iku2* seeds are smaller and that the force needed to perform a
40 µm indentation in *iku2* was almost 20% higher than that required in
the WT. The result of these rupture experiments supports the

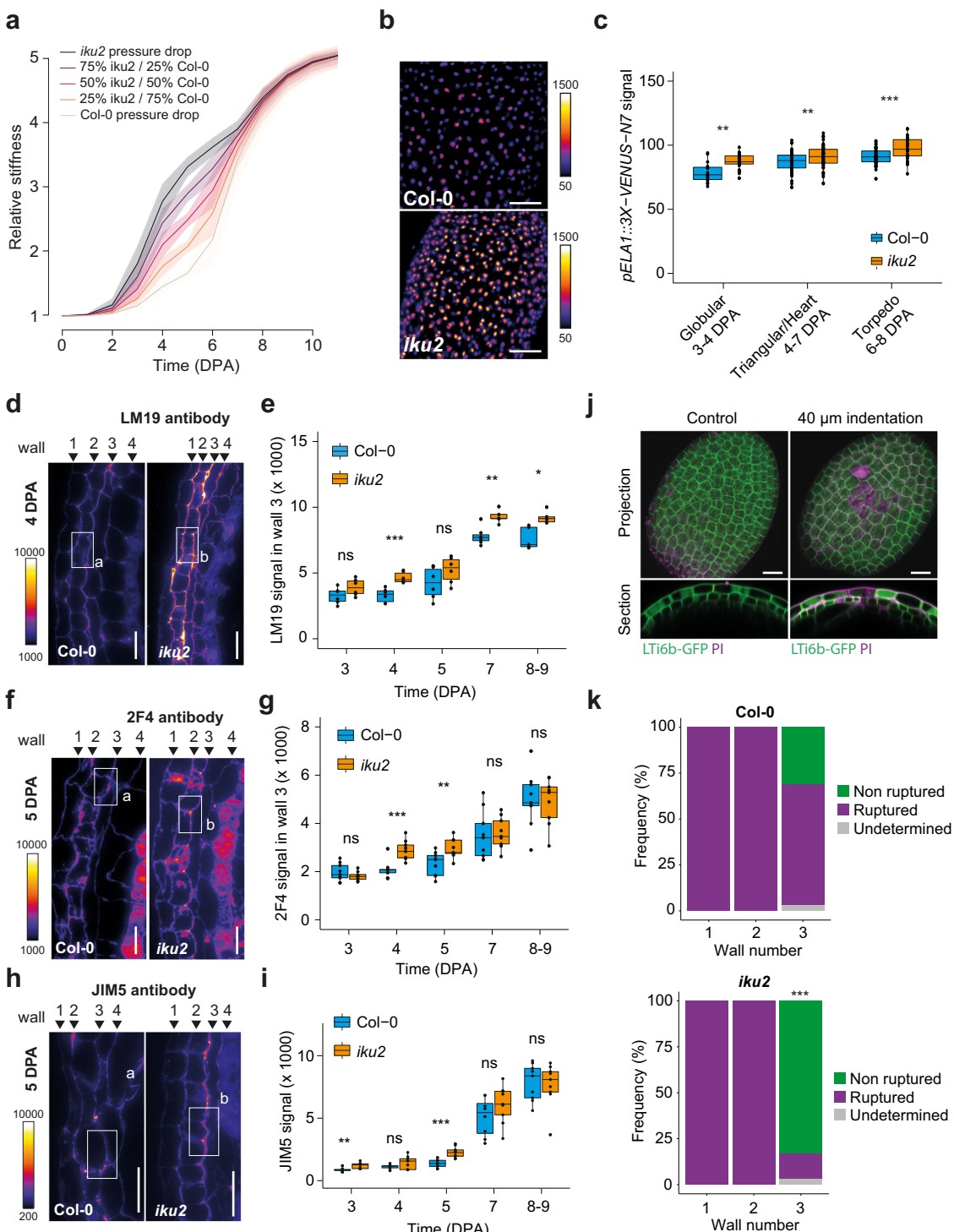

hypothesis that the precocious accumulation of demethylesterified pectins in wall 3 of *iku2* seeds may affect its mechanical properties.

### *iku2* seed growth defects can be rescued by inhibiting testa wall differentiation

To further test the prediction of our model that the reduction of seed growth observed in *iku2* results from precocious testa stiffening, we studied the genetic interaction between *iku2* and a mutant with altered outer-integument identity: *apetala2 (ap2)*. In the testa of *ap2* mutants, the differentiation of the two layers of the outer-integument is impaired, leading to the production of large misshapen seeds[30,31]. We thus analyzed the accumulation of demethylesterified pectins in WT

and *ap2* testa walls. With both LM19 and JIM5 antibodies, we observed a reduction of the fluorescent signal in wall 3 of *ap2* seeds at 6 DPA compared to WT seeds but also noticed an increase in the signal for both antibodies in wall 1 (Fig. 4a–d and Supplementary Fig. 12a, b). These observations support the idea that AP2 activity is, at least partly, necessary for the accumulation of demethylesterified pectins in wall 3. We then combined the *ap2* and *iku2* mutants and quantified the size of the seeds in the double mutants. The seeds of the double mutant *ap2 iku2* were as big as WT seeds but not as big as *ap2* mutant seeds supporting the idea that *ap2* can alleviate some of the growth defects of *iku2* (Fig. 4e and Supplementary Fig. 12c). To investigate this further, we tried to reproduce the results of these experiments in simulations

**Fig. 3 | Testa mechanical properties and response in Col-0 and *iku2*. a** Relative stiffness as a function of time using the drop functions displayed in Fig. 2e as inputs for pressure. All simulations were performed over the 100-parameter value sets best-fitting *iku2* experimental data using *iku2* drop function as an input for pressure. Plain curves display the mean dynamics and confidence intervals show standard deviations. **b** *pELA1::3X-VENUS-N7* expression in Col-0 and *iku2* seeds at the heart embryo stage. Scale bars: 50 μm. **c** Mean signal of *pELA1::3X-VENUS-N7* reporter in nuclei of Col-0 and *iku2* seeds (intensity unit/pixel). Seeds were classified according to embryo developmental stage (one experiment, Col-0, Globular: $n = 15$, Triangular/Heart: $n = 68$, Torpedo: $n = 46$; *iku2*, Globular: $n = 25$, Triangular/Heart: $n = 58$, Torpedo: $n = 30$). Fluorescent signals were compared using two-sided Student tests without adjustments for multiple comparisons, **$p < 0.01$, ***$p < 0.001$. **d, f, h** Signal from LM19 (**d**), 2F4 (**f**) and JIM5 (**h**) immunolocalizations on Col-0 and *iku2* testas at 4-5 DPA. Scale bars: 20 μm. **e, g, i** Signal intensity from LM19 (**e**), 2F4 (**g**) and JIM5 (**i**) immunolocalizations in wall 3 of Col-0 and *iku2* seeds as a function of time (LM19: two independent experiments, Col-0: 3 DPA: $n = 9$, 4 DPA: $n = 9$, 5 DPA: $n = 10$, 6 DPA: $n = 8$, 7–9 DPA: $n = 9$; *iku2*: 3 DPA: $n = 9$, 4 DPA: $n = 9$, 5 DPA: $n = 9$, 6 DPA: $n = 9$, 7–9 DPA: $n = 9$; 2F4: three independent experiments, Col-0: 3 DPA: $n = 9$, 4 DPA: $n = 9$, 5 DPA: $n = 9$, 6 DPA: $n = 9$, 7–9 DPA: $n = 9$; *iku2*: 3 DPA: $n = 8$, 4 DPA: $n = 9$, 5 DPA: $n = 9$, 6 DPA: $n = 9$, 7–9 DPA: $n = 9$, JIM5: three independent experiments, Col-0: 3 DPA: $n = 9$, 4 DPA: $n = 9$, 5 DPA: $n = 10$, 6 DPA: $n = 8$, 7–9 DPA: $n = 9$; *iku2*: 3 DPA: $n = 9$, 4 DPA: $n = 9$, 5 DPA: $n = 9$, 6 DPA: $n = 9$, 7–9 DPA: $n = 9$). Signal intensities were compared using two-sided Student tests without adjustments for multiple comparisons, *$p < 0.05$, **$p < 0.01$, *$p < 0.001$. **j** Maximum projection and z-section of a z-stack of WT seeds at 5 DPA expressing the membrane marker *LTi6b-GFP* (green) and stained with Propidium Iodide (magenta) without indentation or after a 40 μm depth indentation with a conical tip to rupture testa walls. A wall was considered to be ruptured when some intracellular PI staining was found in the underlying cell layer. Scale bars, 20 μm. **k** Quantification of testa wall rupture in Col-0 and *iku2* following a 40 μm indentation with a conical tip. Two independent experiments (Col-0: $n = 35$, *iku2*: 36). Wall rupture frequencies were compared using a two-sided $\chi^2$ test, ***$p < 0.00001$. In all boxplots, the centerline shows the median; the box limits show the upper and lower quartiles; the whiskers correspond to 1.5× interquartile range. Individual points are superimposed in black.

by varying the parameters of the mechanosensitive stiffening function $\{\alpha, \rho, \eta\}$. To do this, we modulated the value of each of these parameters from −20 to +20% in simulations that used either Col-0 or *iku2* pressure drop functions as inputs (Fig. 2e), and analyzed the final relative radius of the seed. We observed that the final seed radius was sensitive to modulations of all three parameters and that $\rho$, the ratio between growth and stiffening thresholds, was the most influential parameter (Fig. 4g and Supplementary Fig. 12d, e). More precisely, we noticed that increasing $\rho$ in Col-0 drop function simulations increased the size of the seed in a similar manner to that observed in the *ap2* mutant; while increasing $\rho$ to similar levels in *iku2* drop function simulations allowed seeds to reach a size comparable to Col-0 seeds, as observed experimentally in the *iku2 ap2* double mutant. Experiments and simulations thus support the hypothesis that the small seed size phenotype of *iku2* is linked to a precocious testa stiffening process that can be partly inhibited in the *ap2* background.

### Osmotic treatments also reveal the antagonistic effects of pressure

Our model and our analysis of the *iku2* mutant phenotype support the hypothesis that endosperm pressure directly promotes growth but indirectly inhibits it through testa stiffening. To further test this antagonistic role of turgor pressure on seed growth, we looked more deeply at the influence of pressure on shell stiffness in simulations. We performed simulations at constant pressure values varying from 0.5× to 1.5× the initial value used in the simulations shown in Fig. 1. For small values of the dimensionless pressure variable (below 0.7), growth was, as expected, either null or limited (Fig. 5a). However, for higher values (above 0.7), increasing pressure induced faster growth initially but led to smaller final radii, as a consequence of a reduction in testa stiffening response time (Fig. 5a and Supplementary Fig. 13). We then tested if this phenomenon could be observed in planta using induced pressure changes.

To do this, we developed a system for in vitro silique culture. We harvested *Arabidopsis* fruits extracted from the plant at 3 DPA and cultivated them in a liquid culture medium for up to 9 days. By sampling seeds at different times, we observed that seeds grew larger in vitro than they did in vivo (Fig. 5b, c). We also noticed that seeds exhibited a small delay in embryo development at 9 DPA (but not at 6 or 12 DPA, Supplementary Fig. 14). As plant hydrostatic pressure builds up from a differential in osmolite concentration between the inside and the outside of the cell[32], we added increasing concentrations of the non-metabolizable sugar sorbitol to the culture medium in order to increase its osmolarity, and thus decrease the hydrostatic pressure of the seed. We observed that seeds sampled from fruits cultivated in media containing high sorbitol concentrations grew more slowly but

for a longer time, and thus became larger, than those sampled from fruits cultivated in media containing little to no sorbitol (Fig. 5c and Supplementary Fig. 14). This effect was reproducible in multiple independent experiments but was weaker than that predicted in simulations (Fig. 5a, c). This discrepancy could again be because we did not take into account the complex structure of the testa in the model. It could also be linked to the fact that plant cells are able to respond to changes in medium osmolarity by altering solute uptake and through metabolic adjustments[33]. Nevertheless, these experiments support both our model and our interpretation of the *iku2* phenotype by showing that endosperm pressure can both promote and restrict seed growth in seeds growing in vitro.

## Discussion
Our results support the hypothesis that endosperm pressure directly promotes growth but indirectly inhibits it due to responses to the tension that it generates in the testa. This tension induces wall 3 stiffening via a mechanism that may involve the accumulation of demethylesterified pectins. Our analyses of the *iku2* mutant further show that the reduction of endosperm pressure that is observed in the WT may not inhibit growth, as previously thought[15], but instead promotes it by delaying force-dependent testa stiffening (Fig. 5d). This interpretation of the *iku2* phenotype is further supported by our osmotic treatments in seeds growing in vitro and by the alleviation of the *iku2* phenotype by the testa differentiation mutant *ap2*.

Our analytic and numerical approaches show that coupling between strain-based growth and stress-based wall stiffening can produce an incoherent feedforward loop that integrates the antagonist effects of endosperm pressure in seed size regulation. In the 1D situation where strain, stress and rigidity are described as scalar quantities, a strain-based stiffening mechanism involving a non-linear response of cell walls to deformation, similar to that proposed to modulate cell growth in the shoot apical meristem[34], could have yielded a similar behavior. However, we chose to consider a stress-based stiffening mechanism for two reasons. First, the literature supports the idea that one of the best characterized mechanosensitive stiffening pathway (i.e., microtubule-dependent cellulose deposition) is more likely to respond to stress than to strain[5,6,35]. Secondly, in a multi-dimensional case where strain, stress and rigidity must be described as tensor fields, we foresee, as have others before us[36], that only a stress-based stiffening mechanism is able to generate anisotropic growth.

How stress and strain are specifically perceived and transduced into biochemical signals remain open questions in plants[36].

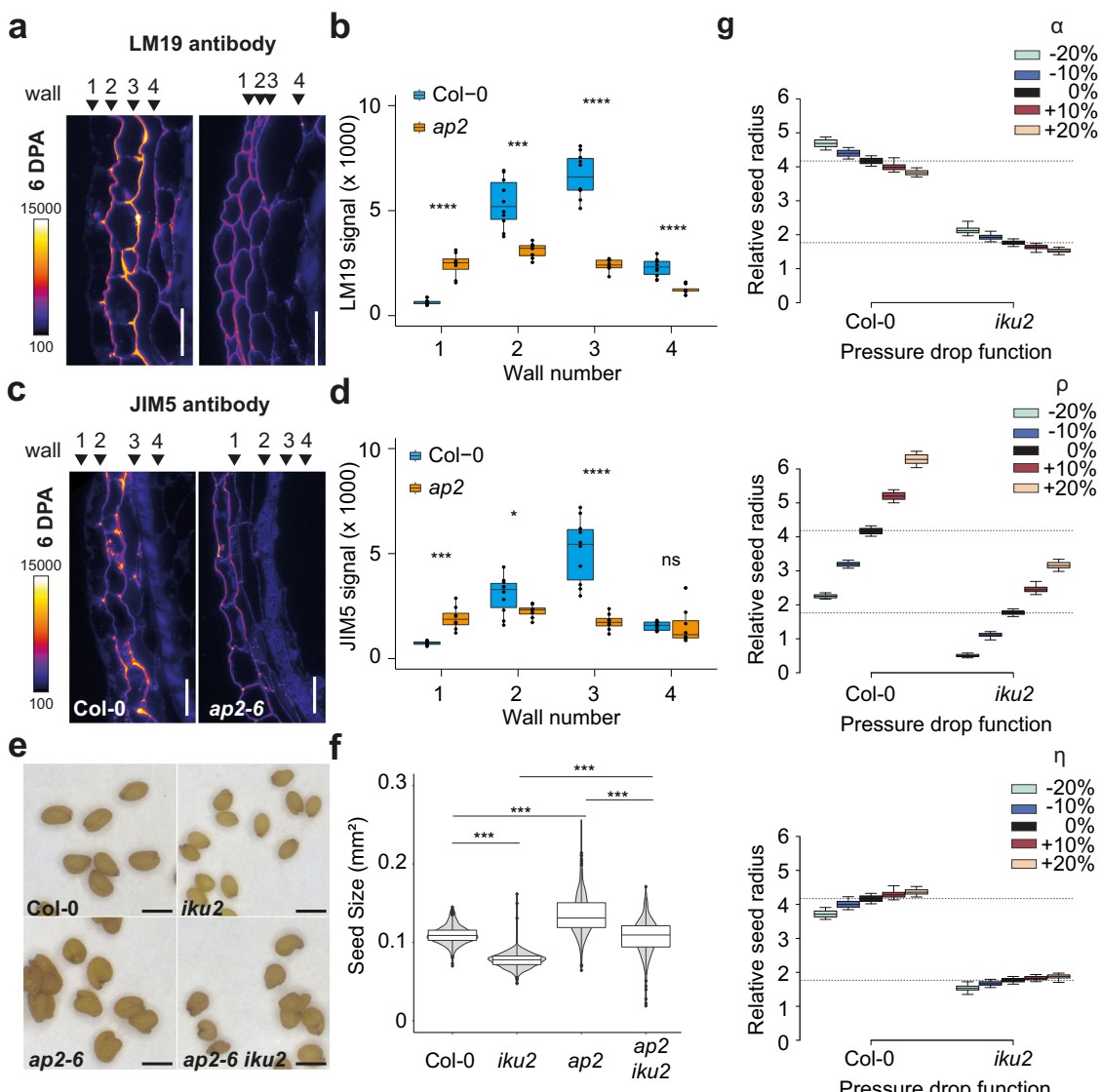

**Fig. 4 | Complementation of the *iku2* phenotype by the mutant of testa identity *ap2*. a**, **c** Labeling of demethylesterified pectins with LM19 (**a**) and JIM5 (**c**) antibodies in testa walls of Col-0 and *ap2* seeds at 6 days post-anthesis. Scale bars: 20 μm. **b**, **d** Quantification of the signal from the LM19 (**b**) and JIM5 (**d**) antibodies in the 4 outermost testa walls of Col-0 and *ap2* seeds (one experiment, Col-0: *n* = 10 seeds, *ap2-6*: *n* = 8 seeds). The centerline shows the median; the box limits show the upper and lower quartiles; the whiskers correspond to 1.5× interquartile range. Individual points are superimposed in black. Signal intensities were compared using two-sided Student tests without adjustments for multiple comparisons, **p* < 0.05, ***p* < 0.01, ***p* < 0.001, *****p* < 0.0001. **e** Representative seeds of Col-0, *iku2*, *ap2-6* and *iku2 ap2-6* at maturity. Scale bars, 0.5 mm. **f** Measurements of seed area in Col-0, *iku2*, *ap2-6* and *iku2 ap2-6* (one experiment: Col-0, *n* = 406; *iku2*,

*n* = 426; *ap2-6*, *n* = 414, *ap2-6 iku2*, *n* = 184). Superimposed on the violin plot, in the boxplot, the centerline shows the median; the box limits show the upper and lower quartiles, the whiskers correspond to 1.5× interquartile range and the single points show outliers. Seed areas were compared using two-sided Student tests without adjustments for multiple comparisons. ****p* < 0.001. **g** Final relative seed radius as a function of three stiffening parameter values (*α, ρ, η*) and of the intensity of the drop of pressure (using either Col-0 pressure drop or *iku2* pressure drop) in the 100 simulations best-fitting *iku2* growth data using *iku2* pressure drop function as an input. In the boxplot, the centerline shows the median; the box limits show the upper and lower quartiles, the whiskers correspond to 1.5× interquartile range and single points show outliers (*n* = 100).

At the molecular scale, forces could be perceived through the deformation of mechanosensitive molecules. How molecular deformations undergone by specific wall components are related to mechanical fields (such as strain and stress) that the wall experiences at the macroscopic scale is another unresolved question. Addressing this will necessitate understanding the nature of the molecular sensors, and analyzing their integration into the wall, and their structural properties. Nevertheless, both theoretical work and experimental observations suggest that cells can indeed perceive and process various types of mechanical signals, which could provide positional and directional cues during development[5,35–37]. We thus predict that mechanosensitive

motives similar to those we have characterized here in developing seeds could be ubiquitous regulators of plant organogenesis.

Finally, our data support the idea that pressure may not simply be a passive force for growth but that it needs to be tightly regulated during organ morphogenesis. This echoes recent work performed in the shoot apical meristem, suggesting that cell hydrostatic pressure may not be homogenous within the epidermis, and may affect growth more locally[38]. Our model also echoes recent work in animals showing that hydrostatic pressure can, together with tissue mechanics, affect embryo growth and size during mouse blastocyst development[39]. In the case of the

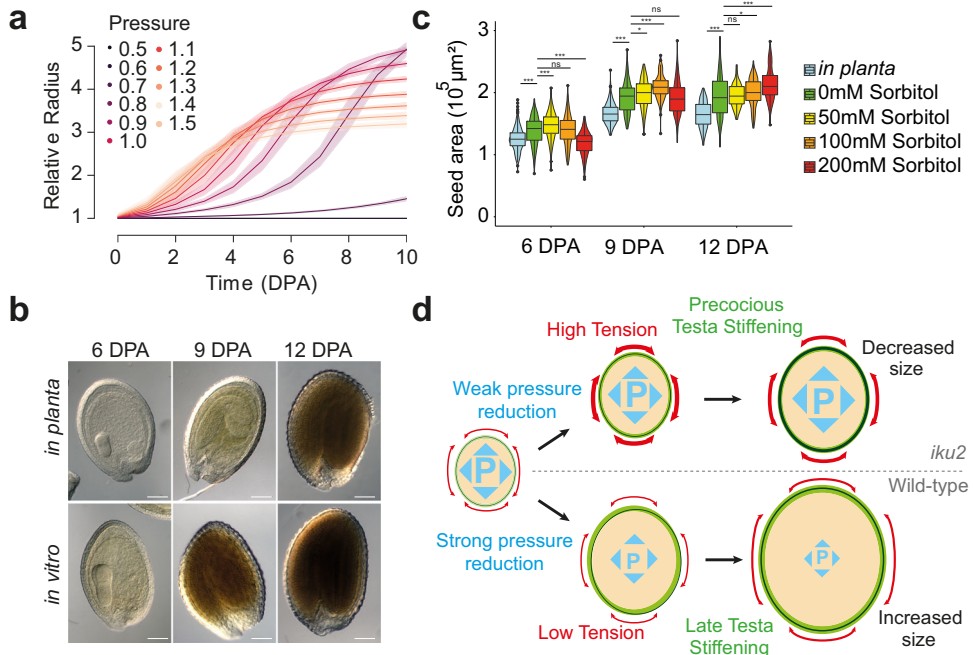

**Fig. 5 | Effect of turgor manipulations on seed growth in an in vitro system.**
**a** Relative radius as a function of time (DPA: Days after pollination) and pressure in simulations. Thick curves and shadowed bands correspond to the mean behavior and standard deviation of the 100 simulations best-fitting WT experimental data using a constant pressure as input. **b** Representative seeds at 6, 9 and 12 DPA from fruits developing in planta or in vitro in culture medium from 3 DPA onward. Scale bars: 100 μm. **c** Measurements of seed areas at 6, 9 and 12 days after pollination from fruits growing in planta or in in vitro culture media with increasing concentration of sorbitol from 3 DPA onwards (two independent experiments; 6 DPA: Uncut, $n = 202$; 0 mM Sorbitol, $n = 184$; 50 mM Sorbitol, $n = 214$; 100 mM Sorbitol, $n = 213$; 200 mM Sorbitol, $n = 198$, 9 DPA: Uncut, $n = 220$; 0 mM Sorbitol, $n = 145$; 50 mM Sorbitol, $n = 227$; 100 mM Sorbitol, $n = 99$; 200 mM Sorbitol, $n = 149$; 12 DPA:

Uncut, $n = 209$; 0 mM Sorbitol, $n = 211$; 50 mM Sorbitol, $n = 161$; 100 mM Sorbitol, $n = 162$; 200 mM Sorbitol, $n = 179$. Superimposed on the violin plots, in all box-plots, the centerline shows the median; the box limits show the upper and lower quartiles, the whiskers correspond to 1.5× interquartile range and the single points show the outliers. Data were compared using two-sided Student tests without adjustments for multiple comparisons, *$p < 0.05$, **$p < 0.01$, **$p < 0.001$, ****$p < 0.0001$. **d** Model integrating the antagonistic roles of endosperm turgor pressure in the control of seed growth. High internal pressure leads to precocious testa stiffening and an early restriction of growth in *iku2* seeds. While in WT seeds, reducing endosperm pressure delays testa stiffening, lengthening the growth phase, which leads to the production of larger seeds.

seed, the counterintuitive effect of pressure on growth could have profound implications for crop improvement strategies, particularly those aiming to alter fluxes of osmotically active metabolites to enhance seed yield[40].

## Methods

### Plant materials and growth conditions

The *iku2-2, ap2-6* and *ede1-3* mutant alleles, and the *LTi6b-GFP* and *pELA1::3X-VENUS-N7* reporters were previously described[11,18,22,41,42]. The *iku2-2* allele was kindly provided by Frederic Berger, the *ede1-3* allele by Claudia Köhler, the *ap2-6* allele by the NASC. The *pIKU2::3X-VENUS-N7* reporter was developed for this study (see below). Seeds were gas sterilized with chlorine (4 mL HCl (37%) in 100 mL bleach) for 2 h and sown on plates with Murashige and Skoog (MS) medium and 0.5% sucrose in sterile conditions, stratified for 2 days at 4 °C and grown for 11 days in a Sanyo (Fisher Scientific) growth cabinet under short-day conditions (8 h light, 21 °C, 150 μmol.m⁻².s⁻¹ during the day, 18 °C during the night). Seedlings were then transferred into separate pots of soil (Argile 10 (Favorit)), and placed in a short-day growth chamber (8 h light, 21 °C and 150 μmol.m⁻².s⁻¹ during the day, 18 °C during the night) for 2–3 weeks before being transferred to a long day growth chamber (16 h light, 21 °C, 150 μmol.m⁻².s⁻¹) to induce flowering. Note that the seedlings were transferred from short day to constant light conditions for the experiments of Supplementary Figs. 5b, d and 6b, c (24 h light, 16 °C, 150 μmol.m⁻².s⁻¹). Seeds were staged every day for up to 10 days by marking the opening of the flower with colored cotton threads.

### Measurements of endosperm turgor by nanoindentation

Siliques were opened and seeds were placed individually on adhesive tape on a microscope slide and covered with water. The slides were then placed on the extended stage of the nanoindenter (TI950 Triboindenter, Hysitron). A truncated conical tip with a flat end of ~100 μm diameter (nominal value = 96.96 μm) was used for indentation. The "displacement-controlled" mode was used to allow imposition of a maximum indentation of 30 μm with a load rate of 6 μm/s (5 s extend, 5 s retract). High-resolution force-displacement curves were recorded with a data acquisition rate of 200 points/s. After indentation, the water was removed and replaced with a drop of clearing solution to allow subsequent embryo staging (see clearing section). Endosperm pressure was calculated using the following formula as described in Beauzamy et al.[15]:

$$F = \frac{\pi P}{c_M} \delta$$

Where $F$ corresponds to the measured force, $P$ to the pressure, $\delta$ to the displacement (indentation), and $c_M$ to the mean curvature of the presumed load-bearing cell wall.

### Extraction of seed mean curvature

The *Lti6b-GFP* membrane marker[42] was used to determine the radius of curvature of Col-0 and *iku2* seeds by confocal microscopy. Individual seeds were placed on adhesive tape on a microscope slide and covered with water. Confocal imaging was performed on a Leica SP8 upright confocal microscope equipped with a 25× water immersion objective (HCX Fluotar VISIR 25×/0.95 W). GFP was excited with a LED laser

emitting at a wavelength of 488 nm (Leica Microsystems). The signal was collected at 495–555 nm for GFP. For large seeds, several z-stacks were taken and stitched with the LAS (Leica Acquisition System) software. The following scanning settings were used: pinhole size 1AE, 1.25× zoom, 15% laser power, 8000 Hz scanning speed (resonant scanner), frame averaging 4–6 times and z intervals of 0.5 μm. After imaging, the water was removed and replaced with a drop of clearing solution to allow subsequent embryo staging (see clearing section). The curvature of the seed was extracted using a custom script developed with the ImageJ software. Seed contours were automatically segmented on "by default" thresholded Z-stack projections (Sum-slices) and rotated to align the ellipse-fitting major axis with the Y-axis. XZ and YZ orthogonal views at the centroid of the seed were displayed and an ellipse was manually drawn to best fit the surface of the turgid compartment of the seed. The radius of curvature was then calculated using RC = (major axis radius)$^2$/(minor axis radius) for longitudinal and transverse curvatures.

### Measurements of testa wall failure by indentation

Col-0 and *iku2* seeds expressing the *LTi6b-GFP* membrane marker were prepared as described for the measurements of endosperm turgor. Nanoindentations were performed with a fluid cell 1 μm 90° conical probe (Ti-0067 03/20/13 (03) Hysitron (Bruker)) on a TI950 Triboindenter (Hysitron). A triangular load function in displacement-controlled mode was applied and the loading rate was set at 10 μm/s. Three indentation depths were tested (30, 40, and 50 μm). Failure of at least one cell wall was visualized on the force curve during the measurement as described in Forouzesh et al.[29]. To assess which wall failed during the indentation, the seeds were incubated for 15 min in a 1 mg/mL Propidium Iodide solution (Sigma) after the indentation, rinsed twice in water and imaged on Leica SP8 upright confocal microscope using similar settings to those used to measure seed curvature. Both GFP and PI signals were acquired. Wall failure was assessed on optical sections obtained using the ImageJ software. A wall was considered as broken if at least one underlying cell accumulated intracellular PI. Only seeds presenting at least one wall pierced in the center were included in the analysis. When the walls were not well defined, the seeds were scored as "undetermined".

### Quantification of *pELA1::3X-VENUS-N7* expression in ad-oi nuclei

The samples were prepared as described in the previous sections. Z-stacks of Col-0 and *iku2* seeds expressing *pELA1:3X-VENUS-N7* were acquired using a Leica SP8 upright confocal microscope equipped with a 40x water immersion objective (HCX APO L UV 40×/0.8 W). After imaging, the water was removed and replaced with a drop of clearing solution to allow subsequent embryo staging (see clearing section). Nuclear Fluorescence intensities were measured using a custom-made macro script developed in ImageJ where the nuclei were segmented using on z-stack projections (Sum-slices) using a marker-based watershed (https://imagej.net/Marker-controlled_Watershed).

### Seed clearing, size measurements and embryo staging

At different days after anthesis, the siliques were opened with a needle, and the seeds were removed with forceps and put in a drop of clearing solution (1 vol glycerol/7 vol chloral hydrate liquid solution, VWR Chemicals) between a slide and a coverslip. The samples were incubated for at least 24 h at 4 °C before being imaged with a Zeiss Axioimager 2 equipped with a 20× DIC dry objective. Seed area was measured by outlining the seed manually using the polygon selection in ImageJ. To compare experimental data with simulations, the radius of the seed was calculated from area measurements by formalizing the seed as a circle (Area = π (radius)$^2$). Seeds were manually classified based on the developmental stage of their embryo. When the embryo was not visible, the seeds were labeled as unclassified. The relative seed growth rate at day (*n*) was calculated using the

following formula: Relative growth rate = (Area (Day$_n$) – Area (Day$_{n-1}$)) / Area (Day$_{n-1}$).

For measurements of seed size at maturity, the seeds were imaged using a Leica stereomicroscope. The resulting images were analyzed using the ImageJ software. The seed outlines were segmented by setting up min and max intensity to 110 and 170, performing a Gaussian Blur of 2px, and using the Make Binary and Analyze Particles functions. The seeds that were too close to each other to be segmented separately were manually removed from the analysis after the segmentation.

### In vitro culture of fruits and sorbitol treatments

Fruits at 3 DPA were cut at the base of the pedicel using a sharp scalpel and immediately placed within a 2 mL Eppendorf tube containing 1.5 mL of liquid medium (1/2 MS (Sigma), 1% Sucrose, 1X Gamborg Vitamins (Sigma), 1/1000 Plant Preservative Medium (Plant Cell Technology)). Once in the tube, the pedicel of the fruit was cut a second time to eliminate the risk of air bubble formation that could block liquid flow in the vasculature. The tubes were kept on top of wet paper in a transparent box that was almost completely closed to reduce evaporation and placed in a growth chamber (16 h light, 21 °C, 150 μmol.m$^{-2}$.s$^{-1}$). Fruits were then harvested at 6, 9 and 12 DPA and the seeds were subsequently cleared and imaged.

### Immunolocalization of cell wall components

Seeds were fixed in ice-cold PEM buffer (50 mM PIPES, 5 mM EGTA and 5 mM MgSO4, pH 6.9) with 4% (w/v) paraformaldehyde. The samples were placed under vacuum (2 × 30 min on ice), rinsed twice in PEM buffer, dehydrated through an ethanol series and infiltrated with increasing concentrations of LR White resin in absolute ethanol (London Resin Company) over 8 days before being polymerized at 60 °C for 24 h. The samples were sectioned (1.0 μm thickness) using a diamond knife 45° angle (Diatome, LFG Distribution) mounted on a Leica RM6626 microtome and dried onto glass slides. For JIM5 and LM19 antibodies (Plant Probes), the sections were initially blocked in a PBS solution with 3% (w/v) BSA for 1 h at room temperature. For the 2F4 antibody (Plant Probes), the sections were initially blocked in TCaS buffer (20 mM Tris-HCl, pH 8.2, 0.5 mM CaCl$_2$, 150 mM NaCl) with 3% (w/v) skimmed milk for 1 h at room temperature. The antibodies were applied to the sections overnight at 4 °C in a humid chamber. The JIM5 and LM19 antibodies were diluted 1:10 (v/v) in PBS/BSA 1% while the 2F4 antibody was diluted 1:5 in 1% skimmed milk in TCaS buffer. The sections were then washed in excess of the buffer to dilute the antibody and subsequently incubated for 1 h at room temperature with the secondary antibody (anti-rat IgG Alexa 488, anti-rat IgM Dylight Alexa 488, and anti-mouse IgG Alexa 488 for JIM5, LM19 and 2F4 respectively) diluted 1:100 in the same buffers as those used for diluting the primary antibody. The sections were washed in buffer solutions as described above and covered with PBS or with TCaS buffer. The samples were then counterstained with filtered Calcofluor White M2R (fluorescent brightener 28; Sigma-Aldrich) at 0.25 mg.mL$^{-1}$ and mounted with VECTASHIELD (Eurobio). The sections were imaged using a Zeiss Axioimager 2 equipped with a 40× dry objective.

Quantification of the immunofluorescence signal was performed with a custom macro script developed on the ImageJ software. The two channels were split; the first channel was labeled as the "control" (Calcofluor), and the second channel as the "signal" (antibody) (Supplementary Fig. 7). Testa cells were segmented from the control channel using a stationary wavelet transformation and a marker-based watershed (https://imagej.net/Marker-controlled_Watershed). Each testa cell was then manually assigned to its layer. Enlarged region of interest (ROI) for a given cell (layer *n*) and for its neighboring cells (layer *n*−1) were added onto a new image and used to define cell wall junctions as being the common region between the two (ImageCalculator (And...) Command). The newly defined ROI was then transferred to the Signal channel for intensity measurements. Finally,

cell wall ROIs were overlaid onto the composite image of control and signal channels to manually check that the segmentation and localization of the walls had been correctly performed.

For toluidine blue staining, the sections were incubated for 20 s at 70 °C with filtered Toluidine Blue 1%/1% borax before being rinsed with distilled water, dried and mounted in Entellan mounting medium (Merck). The sections were imaged with a Zeiss Axioimager 2 equipped with a 20× dry objective.

### Generation of the *pIKU2::3X-VENUS-N7* line

The *IKU2* promoter was amplified using the primers Prom-IKU2-B4 and Prom-IKU2-B1R (Supplementary Table 7) and cloned into *pDONR-P4-P1R* (Life Technologies). A triple LR Gateway reaction (Life Technologies) was then performed using the *pIKU2-pENTR-R4-L1*, *3X-VENUS-N7-pENTR-L1-L2*, and *3'-ter-pENTR-R2-L3* plasmids as entry vectors and the *pH7m34GW* plasmid as destination vector to generate a *pIKU2::3X-VENUS-N7-pH7m34GW* construct (conferring Hygromycin resistance in plants).

### Genotyping

The *iku2-2* mutant allele was genotyped through its deletion using the primers: iku2-Del-For and iku2-Del-Rev (Supplementary Table 7). The *ap2-6* allele was selected based on the *ap2* flower phenotype and verified by PCR and sequencing using primers initially designed to incorporate AP2 CDS using the Gateway system (Supplementary Table 7).

### Quantitative gene expression analysis

Total RNA from siliques at 2 and 5 DPA was extracted using the Spectrum Plant Total RNA Kit (Sigma). Total RNAs were digested with Turbo DNA-free DNase I (Ambion) according to the manufacturer's instructions. The mRNAs were reverse transcribed using the SuperScript VILO cDNA Synthesis Kit (Invitrogen) according to the manufacturer's protocol. PCR reactions were performed in an optical 384-well plate in the QuantStudio™ 6 Flex Real-Time PCR System (ThermoFisher Scientific), using FastStart Universal SYBR Green Master (Rox) (Roche), in a final volume of 10 μL, according to the manufacturer's instructions. The following standard thermal profile was used for all PCR reactions: 95 °C for 10 min, 40 cycles of 95 °C for 10 s, and 60 °C for 30 s. The data were analyzed using the QuantStudio Real-Time PCR Software v1.3 (Applied Biosystems). As a reference, a geometric mean between two house-keeping genes, *EIF4A1* and *AP2M*, was used to normalize *ELA1* expression. For each couple of primers, PCR efficiency (E) was estimated from the data obtained from standard curve amplification using the equation $E = 10^{-1/\text{slope}}$. Expression levels are presented as $E^{-\Delta Ct}$, where $\Delta Ct = Ct_{GOI} - Ct_{REF}$. The sequence of the primers used for qPCR can be found in Supplementary Table 7.

### Statistical analysis

No sample size calculation was performed prior to the experiments. The number of seeds (biological replicate) analyzed in each experiment was chosen given technical constraints (in time and material) specific to each experiment. At least, three seeds harvested from three different siliques were used per independent experiment. Each experiment was carried out independently at least twice (i.e., from independent batches of plants growing at different times). Data from independent experiments were pooled except in the following cases: (1) when the growth conditions of the independent experiments were different (long days or continuous light, see Plant material and growth conditions section), (2) when the settings used for the acquisition of fluorescent images were not the same, and (3) where sample sizes were very different between independent experiments. In the experiments involving imaging by optical and confocal microscopy, seeds that were strongly injured during sample preparation were excluded from the analysis. In the piercing experiments, only seeds presenting at least one wall pierced in the center were included. No randomization or blinding was performed. The data were analyzed using Excel (v. 2016), the R software (v.4.2.0), Python (V.3.7.5) or Jasp (v.0.16.3). When experimental data were compared, notably using Student tests, statistical significance was displayed with stars with the following nomenclature: $*p < 0.05$, $**p < 0.01$, $***p < 0.001$ and $****p < 0.0001$. Exact $p$ values can be found in the Source Data file. In boxplot representations, the midline represents the median of the data while the lower and upper limits of the box represent the first and third quartile respectively. The bars represent the distance between the median and one and a half times the interquartile range. When the number of biological repeats was low (for pressure measurements, the analysis of *ELA1* expression and the immunolocalizations), individual measurements were superimposed as points on the boxplots. For the remaining representations, points, often connected with lines, correspond to the mean and the error bars to the standard deviation.

### Theoretical modeling and numerical simulations

Details about the assumptions, derivation and implementation of our model are gathered in a document provided in Supplementary Notes 1–3. The first section details the underlying assumptions and the dimensionless formalization; the second section explains the steady state analysis we performed and the third section presents its numerical implementation.

### Reporting summary

Further information on research design is available in the Nature Portfolio Reporting Summary linked to this article.

## Data availability

The experimental data generated for this article are available on Zenodo and can be found following this link: https://doi.org/10.5281/zenodo.7078846. They can also be found on the server of the laboratory of Plant Development and Reproduction (https://flower.ens-lyon.fr/). The experimental measurements used to generate the figures of this article are provided in the Source Data file. Source Data are provided with this paper.

## Code availability

The code used in this study is available at https://gitlab.inria.fr/mosaic/publications/seed_sup_mat.

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

## Acknowledgements

We thank Frederic Berger and Claudia Köhler for providing the *iku2-2* and *ede1-3* mutant seeds, respectively; Romain Azais, Florian Ingels, Yuchen Long, Arezki Boudaoud, Christophe Godin and Oliver Hamant for helpful discussions and comments on the manuscript; Lena Beauzamy and Simone Bovio for technical assistance regarding indentation; Guillaume Cerutti and Jeremy Just for technical assistance regarding the distribution of the data and the code; Alexis Lacroix, Patrice Bolland and Justin Berger for technical assistance regarding plant cultivation; Isabelle Desbouchages and Hervé Leyral for technical assistance regarding molecular biology work; Cindy Vial, Laureen Grangier, Nelly Camilleri and Stéphanie Maurin for administrative assistance; the PLATIM (SFR128 Biosciences) for technical assistance regarding the microscopy. This study was financed by the research fund of the ENS de Lyon (France) and by the BAP department of INRAE (France).

## Author contributions

O.A., G.I. and B.L. led the study, obtained funding and supervised the work. A.C., C.B. and B.L. carried the experiments. A.C., C.B., V.B. and B.L. analyzed the experiments. O.A. developed the theoretical model and performed the numerical simulations. O.A., G.I. and B.L. wrote the paper with input from all authors.

## Competing interests

The authors declare no competing interests.
