## [Peer Review File · Nature Communications]

Evidence that endosperm turgor pressure both promotes and restricts seed growth and sizeReviewer #1 (Remarks to the Author):

In this manuscript, the authors use experimental and modeling approaches to test the hypothesis that pressure-induced stresses play two antagonistic roles in the control of seed development. On one side, those stresses are proposed to directly drive seed growth via the increase in turgor pressure in the endosperm. On the other side, they also indirectly inhibit the growth via mechanosensitive stiffening of the seed coat walls. Authors suggest that the progressive decrease in turgor pressure during normal development is necessary to ensure the proper growth of the seed. The partial impairment of growth in *iku2* mutant is suggested to be driven by prolonged maintenance of higher turgor pressure that leads to precocious stiffening of the seed coat via mechanosensitive induction of pectin modifications. This is a very interesting new mechanism that could potentially control the cessation of growth in various systems. The Paper is well written and most of the data support the main hypothesis. However, I have some major concerns about some experiments:

105-107: "To our surprise, this decrease was not observed in the *iku2* mutant, where endosperm pressure was constant and higher than in the WT from the globular stage onward (3 to 4 DPA)" – The behavior of WT and *iku2* is variable and differs between 3 independent replicates of the turgor pressure measurements. Is the turgor pressure really constant in *iku2* over time? It looks like the values are decreasing in both samples provided in extended Fig 4c. This decrease seems to be very similar to the WT in replicate 2. Additionally, in replicate 2 (at torpedo stage) the pressure is statistically higher in the WT as compared to *iku2*. How do authors explain this inconsistency? Based on the data provided, I don't think one can conclude that: "This decrease was not observed in the *iku2* mutant, where endosperm pressure was constant and higher than in the WT from the globular stage onward". In my opinion, the proposed model is not supported by currently provided data on the dynamics of turgor pressure.

Other points:

Why is the inner wall of layer 3 so wavy? If this is the suggested load-bearing element to sustain the pressure of the endosperm, one would expect it to be stretched upon this pressure. Instead, those walls seem to be bulging toward the inside of the developing seed (e.g. Fig. 3D, extended Data Fig 7d, 8b, 9b).

Strain stiffening was suggested to play a role in limiting the growth of the central zone of the shoot apical meristem (Science 335, 2012). How do the findings here relate to those data? Could strain stiffening be also involved in the control of seed size? I would suggest discussing this in the paper.

99-101: "Growth rate was initially higher in *iku2* seeds than in the WT, but decreased faster, ultimately leading to the production of smaller seeds (Fig. 2a-b, Extended Data Fig. 4b)." – Statistical analysis is needed to confirm this statement.

The main Fig 3C shows data from 2 independent experiments combined and Extended Fig 6a from one additional experiment. Either show graphs from all 3 independent experiments combined or show 3 graphs for each experiment independently.

Fig 3B: Color scales range from "+" to "-". In the adjacent graph (in C) there are A.U. quantified. I suggest that the A.U. range is indicated in B. The same comment applies to all other images where heat maps of signal intensity are shown.

Reviewer #2 (Remarks to the Author):

In this manuscript Creff et al expand on their previous study published in the same journal that elegantly showed that seed size was controlled by the interplay between endosperm turgor, and turgor-stimulated stiffening of a specific cell layer in the seed coat. This model was counter to our understanding at the time and set the standard for our understanding of seed size control.

Here the same authors go one step further and show using a model, parameterized using turgor

and stiffness measurements in seed size mutants, and confirm that their measurements are consistent with their previous conclusions. Perhaps most striking is the discovery that *iku2* mutants have high turgor rather than low turgor as previously hypothesized, but the authors also discover further features of seed size mutants that are consistent with the conclusions of their previous paper.

That the model also works is interesting but perhaps less important in that the behavior of simple deterministic models of the type described are well known, and fairly obvious. Perhaps the best use of the model is to show that measured changes in turgor between the WT and *iku2* are sufficient to generate a fall in seed size.

So my assessment is that the study is very interesting, and essentially provides further evidence to confirm the authors theory for seed size control. These are the key strengths.

To my mind the only important weakness is the conspicuous absence of any new genetic data to support the conclusions. It seems obvious to cross *iku2* to the ELA1 amiRNA lines generated in the 2015 paper and show that this restores seed size, or indeed to other mutants which more strongly affect testa strengthening such as *ap2*. If the authors have tried this they should report the results or explain why they haven't (in case there is a reason i miss). The paper would be much stronger with this additional data.

Minor point: there is a lot of interesting data in the paper but very little of it in figure 1. This could be reconsidered.

Reviewer #3 (Remarks to the Author):

The authors are interested in the question of size regulation of the arabidopsis seed. It is an interesting and important problem, both for the understanding of development and its practical applications. They develop a simple model representing the seed as a spherical shell, pressurized by turgor. The motivation for this is that the testa is thought to restrict growth of the inner endosperm. Growth is strain-based, which means it directly depends on the stiffness of the shell (testa). They present some data that suggests that demethylation and subsequent cross-linking of pectin is involved in the stiffness increase that causes growth to stop, determining the final seed size.

Some aspects of the model make sense. For example the idea that the testa restricts growth, and that it becomes stiffer in later stages as growth stops. In their model, the shell becomes stiffer when it is more stressed, and this stiffness increase leads to growth cessation. They find that the model parameters that best match the data are when the stiffness response to stress occurs late (just before growth stops) and if it is highly non-linear, so essentially acting like a switch. The stress in a sphere is directly proportional to the radius, so the growth stops when it reaches a certain size. Is such a model plausible? It could be if there was a way to measure stress. Unfortunately, there isn't any direct way to measure stress, it has to be inferred by looking at the deformation of a material of known stiffness. Like a scale, you can measure the weight(force) by looking at the deflection of a spring of known stiffness.

Although it is possible for a cell to measure strain, and potentially deduce stress from that, it is unclear how that would work in a system that is also growing, and how the plant can distinguish strain from growth from elastic strain. This is an unsolved problem for all models that posit stress as a signalling mechanism for the development of growing plant organs.

The authors then suggest that such a model could explain why WT seeds grow larger than the *iku2* mutant, which when measured with indentation, appears to have higher turgor pressure. In a stress based model, lower pressure means lower stress, and a larger seed. Another explanation could be that the plant arrests growth early, and the increased pressure is just a response to that. There are no experiments to test if the pressure really is the determining factor there. I think such a claim required a bit more evidence in that regard.

Specific comments:

Abstract: I don't think the term "incoherent feed forward loop" is correct in this context, it will probably confuse rather than assist most readers in understanding the model. I would recommend removing it as it does not really add anything.

Specific comments

Lines 35: Considering the limitations of the work in reference (10), I am not sure that can be generally accepted as fact.

Lines 67: Why? What is the purpose of seeing if the process can be called an IFFL? Why does it matter?

Line 70: "assimilated to a", perhaps "approximated as"

Line 75: The model should be better described here. The reader should not have to go to the supplement, a high level description of the model needs to be added here. At the minimum it should mention what the growth and the stiffening depend on. In the end the model is very simple, it stops growing at a certain stress, and stress directly depends on size and pressure.

Lines 78: The authors need to explain why the growth stops (if always true). Is that necessarily true for all parameters? For example, it is true that for parameters found in ii) where the growth doesn't stop, that lowering the pressure cannot change the outcome?

Lines 79-80: non-sequitur, I don't see how this follows. If it said reached a steady state at the same size, then this would make sense.

Lines 81-90: If you make the cell stiffer based on stress, and stress is proportional to pressure and size, it makes sense that this should follow. Again, this should be explained.

Lines 92-94: This is a bit problematic. A strongly non-linear response based on stress that happens late means the model is programmed to stop when it hits a certain stress. Since stress is directly proportional to size, the model really just stops growing when it hits a certain size. This would require that the cell have a method of sensing stress (independent of strain) which is not physically possible. There is no method to measure stress that does not involve strain.

Lines 112-128: If growth essentially shuts down when a threshold stress is reached, then lower pressure would give larger seeds, as stress is an increasing function of both pressure and size. An alternate hypothesis is that growth is simply reduced in the mutant, and that pressure is increased because the plant tries to fight that.

Lines 137: Doesn't this argue that pressure is increased as a response to early stiffening?

What happens in the *ela* mutants? I guess the plants are bigger so the seeds are probably bigger as well, but does that also mean the whole plant is size controlled by a similar mechanism? Leaves, stems, etc. Seems unlikely. What about polyploids?

Lines 147: I would say that the evidence supporting the idea that PME_s promote growth is pretty weak.

Lines 163-176: The authors state that they want to see if the testa is stiffer, and test this by determining the failure point during indentation. This does not probe stiffness, but rather strength of the layer. They say they observe more force for *iku2* seeds than WT at all indentation depths, although no force-indentation curves are shown. They then say that this could represent turgor pressure or stiffness differences, although the former would seem more likely, since they are doing rather large indentations. They do not say if they indent seeds of a similar size, so if the *iku2* is

both smaller and stiffer, that could indicate significantly higher turgor pressure, since a larger structure should appear stiffer at the same pressure. In any case these experiments don't really address what is suggested in the opening sentence of this section. The authors might consider to do osmotic treatments on the seeds. This could give an indication of the pressure difference. Indentation on plasmolyzed seeds could also indicate if bending stiffness of the testa is a factor for the larger observed indentation stiffness in iku2.

Supplement:

Equation SE2: At this point it might be good to mention that this equation applies to the deformed configuration, that is after it is pressurised.

Text after SE3: I don't think the Lockhart or Ortega models were developed for FEM analysis. Ref (4) Boudon et al. 2015 is almost the same model as Bassel et al. 2014, PNAS.

Cell wall Stiffening section: "de degradation term".

REVIEWER COMMENTS

Reviewer #1 (Remarks to the Author):

In this manuscript, the authors use experimental and modelling approaches to test the hypothesis that pressure-induced stresses play two antagonistic roles in the control of seed development. On one side, those stresses are proposed to directly drive seed growth via the increase in turgor pressure in the endosperm. On the other side, they also indirectly inhibit the growth via mechanosensitive stiffening of the seed coat walls. Authors suggest that the progressive decrease in turgor pressure during normal development is necessary to ensure the proper growth of the seed. The partial impairment of growth in *iku2* mutant is suggested to be driven by prolonged maintenance of higher turgor pressure that leads to precocious stiffening of the seed coat via mechanosensitive induction of pectin modifications. This is a very interesting new mechanism that could potentially control the cessation of growth in various systems. The Paper is well written and most of the data support the main hypothesis.

We thank the reviewer for this positive assessment of our paper.

However, I have some major concerns about some experiments:

105-107: “To our surprise, this decrease was not observed in the *iku2* mutant, where endosperm pressure was constant and higher than in the WT from the globular stage onward (3 to 4 DPA)” – The behavior of WT and *iku2* is variable and differs between 3 independent replicates of the turgor pressure measurements. Is the turgor pressure really constant in *iku2* over time? It looks like the values are decreasing in both samples provided in extended Fig 4c. This decrease seems to be very similar to the WT in replicate 2. Additionally, in replicate 2 (at torpedo stage) the pressure is statistically higher in the WT as compared to *iku2*. How do authors explain this inconsistency? Based on the data provided, I don't think one can conclude that: “This decrease was not observed in the *iku2* mutant, where endosperm pressure was constant and higher than in the WT from the globular stage onward”. In my opinion, the proposed model is not supported by currently provided data on the dynamics of turgor pressure.

We agree with the reviewer that there is some variability in our measurements of turgor pressure (both within and between experiments). We performed and showed three independent replicates of this experiment so that the readers could actually see this variability. This variability is not surprising and was already observed in the article of Beauzamy and colleagues describing the method. It could result from the fact that we measure pressure indirectly from indentations. However, the article of Beauzamy and colleagues also shows that more direct measurements of endosperm pressure with a pressure probe also show high variability between samples. It is thus plausible that the variability we observe is not only technical and that the pressure within the endosperm could vary from seed to seed, maybe because of changes in plant hydration or in response to intrinsic or environmental signals. As shown in Supplementary Figure 1b, the WT seed growth pattern is also rather variable from experiment to experiment, which would support the influence of extrinsic, but as yet unidentified, factors on our system.

Regarding the trend in the *iku2* mutant, it is true that the experiment shown in the main text (which is the one we used to fit pressure in simulations to the experimental data) does not show any decrease in pressure over time in the mutant but that there is a small trend in the experiments shown in the supplementary figures. However, this decrease is always smaller than that observed in WT. As we know show in Supplementary Fig.5d, we should still have higher growth in the WT even if pressure also reduces in *iku2* (as long as pressure is reduced more in WT than in the *iku2* mutant, something that we see in all repeats). Nevertheless, we corrected the manuscript according to the reviewer's comments and now state:

*Lines 140-142: “To our surprise, this decrease was less pronounced, if not nonexistent, in *iku2* mutants so that *iku2* pressure was generally more elevated than that in the WT during most of the growth phase (from 3 to 4 to 7 DPA).”*

Regarding the last point (Torpedo stage) of replicate 3, it is indeed the only case where we see that pressure is statistically higher in the WT than it is in *iku2*. We do not know why, it could be due to an environmental fluctuation. However, we did not focus on this point because growth has almost terminated at the torpedo stage (as shown in Supplementary Fig. 1b-c). As a result, even if pressure were to increase at this stage, it would not significantly affect our system as it has already almost reached steady state.

Other points:

Why is the inner wall of layer 3 so wavy? If this is the suggested load-bearing element to sustain the pressure of the endosperm, one would expect it to be stretched upon this pressure. Instead, those walls seem to be bulging toward the inside of the developing seed (e.g. Fig. 3D, extended Data Fig 7d, 8b, 9b).

The waviness of the inner wall of layer 3 that we see in the immunolocalization is an artefact of the fixation and sectioning. As we can see from the picture below showing a longitudinal section of a seed expressing the membrane marker *LTi6b-GFP* at 5DPA, wall 3 is not so wavy in living, turgid seeds.

Strain stiffening was suggested to play a role in limiting the growth of the central zone of the shoot apical meristem (Science 335, 2012). How do the findings here relate to those data? Could strain stiffening be also involved in the control of seed size? I would suggest discussing this in the paper.

Strain-stiffening is a material property found in many biopolymers. Under intense mechanical loading, the stress-strain relationship becomes nonlinear and the material's effective stiffness increases. It has indeed been nicely shown by Kierzkowsky and colleagues that such strain stiffening could affect growth in the shoot apical meristem. By definition, strain-stiffening is a second order mechanical property (it is necessary to go beyond the first-order linear model of elasticity to consider it). In the present work, it proved unnecessary to explore such higher-order rheological properties to account for our experimental observations. We therefore concentrated on a parsimonious linear and homogeneous elastic model. It is important to note that we considered a first-order geometrical model of the seed (a homogeneous sphere with no differences between regions). In a more elaborate model, where seed profiles are not restricted to spherical caps, such mechanisms could very well be involved in the regulation of shape anisotropy. However studying these mechanisms is beyond the scope of this study. Nevertheless, our mechanosensitive stiffening process could be assimilated to strain stiffening, as the stiffness of our system increases with the mechanical load applied to it. However, mechanistically, our system is different from that described by Kierzkowsky et al as it does not rely on non-linear material properties but on active mechanical signaling which affects cell wall properties (potentially through the synthesis or remodeling of cell wall components). Nevertheless, as reviewer 2 suggested, we added the following sentence to the discussion:

Lines 259-263: While strain stiffening, a mechanism involving a non-linear response of cell walls to deformation, has been shown to regulate cell growth in the shoot apical meristem, our analytic and numerical approaches show that a coupling between strain-based growth and stress-based wall stiffening can produce an incoherent feedforward loop that can explain the dynamics of seed growth and size regulation.

99-101: "Growth rate was initially higher in *iku2* seeds than in the WT, but decreased faster, ultimately leading to the production of smaller seeds (Fig. 2a-b, Extended Data Fig. 4b)." – Statistical analysis is needed to confirm this statement.

This observation was based on the fact that *iku2* ovules were slightly smaller than WT ovules but that seeds at 1 and 2 DPA were statistically larger. However, this effect is not strong enough to allow statistical support when comparing growth rate differences as this parameter (which is calculated by deriving successive measurements of seed size) is very variable (as observed in Supplementary Fig. 5A). We have thus changed our text and now state:

*Lines 134-136 "We observed that *iku2* seeds initially grew similarly to WT seeds but that after a few days, growth decreased faster in *iku2* than it did in the WT, ultimately leading to the production of smaller seeds (Fig. 2b-c, Supplementary Fig. 5a)."*

The main Fig 3C shows data from 2 independent experiments combined and Extended Fig 6a from one additional experiment. Either show graphs from all 3 independent experiments combined or show 3 graphs for each experiment independently.

We thank the reviewer for pointing out this mistake. Fig.3C actually displays the result of a single independent experiment. As the settings and growth conditions used were slightly different between the two independent experiments, we did not pool them even though the result was similar. Consequently, one experiment is shown in the main figures and one in supplementary data.

Fig 3B: Color scales range from "+" to "-". In the adjacent graph (in C) there are A.U. quantified. I suggest that the A.U. range is indicated in B. The same comment applies to all other images where heat maps of signal intensity are shown.

Based on this comment we modified all of the confocal pictures and corresponding graphs measuring fluorescent signals in the manuscript so that the raw intensity is shown rather than "+" and "-" and A.U.

Reviewer #2 (Remarks to the Author):

In this manuscript, Creff et al expand on their previous study published in the same journal that elegantly showed that seed size was controlled by the interplay between endosperm turgor, and turgor-stimulated stiffening of a specific cell layer in the seed coat. This model was counter to our understanding at the time and set the standard for our understanding of seed size control.

Here the same authors go one step further and show using a model, parameterized using turgor and stiffness measurements in seed size mutants, and confirm that their measurements are consistent with their previous conclusions. Perhaps most striking is the discovery that *iku2* mutants have high turgor rather than low turgor as previously hypothesised, but the authors also discover further features of seed size mutants that are consistent with the conclusions of their previous paper. That the model also works is interesting but perhaps less important in that the behaviour of simple deterministic models of the type described are well known, and fairly obvious. Perhaps the best use of the model is to show that measured changes in turgor between the WT and *iku2* are sufficient to generate a fall in seed size.

So my assessment is that the study is very interesting, and essentially provides further evidence to confirm the authors theory for seed size control. These are the key strengths.

To my mind the only important weakness is the conspicuous absence of any new genetic data to support the conclusions. It seems obvious to cross *iku2* to the *ELA1* amiRNA lines generated in the 2015 paper and show that this restores seed size, or indeed to other mutants which more strongly affect testa strengthening such as *ap2*. If the authors have tried this they should report the results or explain why they haven't (in case there is a reason i miss). The paper would be much stronger with this additional data.

We thank the reviewer for assessing our work and for this suggestion. We did not show any new data using the *pELA1::ELAamiRNA* described in the 2015 paper because we have problems of silencing so that the seed size phenotype of these lines seems to be less and less strong over generations. However, as the reviewer suggested, we added new data on the *ap2* mutant. In Fig. 4, we now show that *ap2* seeds have defects in demethylesterified pectin accumulation in wall 3 (shown using JIM5 and LM19, but not 2F4 because this antibody is no longer commercialized). These defects are logical given that AP2 is known to control outer-integument differentiation, and they correlate with the production of larger seeds, which is consistent with our model. As requested by the reviewer, we also show that the small seed size phenotype of the *iku2* mutant phenotype can indeed be attenuated by removing AP2 function, further supporting our interpretation of the *iku2* phenotype.

Minor point: there is a lot of interesting data in the paper but very little of it in figure 1. This could be reconsidered.

We now added a new panel in Fig.1 to better explain the relevance of our model, as also suggested by reviewer 3.

Reviewer #3 (Remarks to the Author):

The authors are interested in the question of size regulation of the arabidopsis seed. It is an interesting and important problem, both for the understanding of development and its practical applications. They develop a simple model representing the seed as a spherical shell, pressurized by turgor. The motivation for this is that the testa is thought to restrict growth of the inner endosperm. Growth is strain-based, which means it directly depends on the stiffness of the shell (testa). They present some data that suggests that demethylesterification and subsequent cross-linking of pectin is involved in the stiffness increase that causes growth to stop, determining the final seed size.

Some aspects of the model make sense. For example the idea that the testa restricts growth, and that it becomes stiffer in later stages as growth stops. In their model, the shell becomes stiffer when it is more stressed, and this stiffness increase leads to growth cessation. They find that the model parameters that best match the data are when the stiffness response to stress occurs late (just before growth stops) and if it is highly non-linear, so essentially acting like a switch. The stress in a sphere is directly proportional to the radius, so the growth stops when it reaches a certain size. Is such a model plausible? It could be if there was a way to measure stress. Unfortunately, there isn't any direct way to measure stress, it has to be inferred by looking at the deformation of a material of known stiffness. Like a scale, you can measure the weight(force) by looking at the deflection of a spring of known stiffness.

Although it is possible for a cell to measure strain, and potentially deduce stress from that, it is unclear how that would work in a system that is also growing, and how the plant can distinguish strain from growth from elastic strain. This is an unsolved problem for all models that posit stress as a signalling mechanism for the development of growing plant organs.

We thank the reviewer for his assessment of our work and his helpful comments and suggestions.

We agree with the reviewer that our manuscript, like others positing stress as a signaling cue, is indeed limited by the fact that we do not know how stress within the wall is sensed. The lack of molecular mechanisms for mechanoperception in plants is currently a limitation for the entire field of plant biomechanics. However, many studies still suggest that cells are able to sense stress and that it could be a key factor controlling morphogenesis. The usefulness of specific stress- and strain-sensitive signaling pathways in morphogenesis has been highlighted by others in the past (Bozorg et al, PLoS Comp. Biol. (2014)). Our model can be seen as further evidence for the utility of such diversity in the cellular mechanosensitive modalities. The fundamental difference between strain and stress within this study is that while stress relies only on the geometry of the considered surface and the inner pressure (here $=pr$), strain also depends on the elastic and structural properties of the testa cell walls through their effective rigidity ($=pr/k$). Multiple experimental indicators have been reported suggesting that while irreversible cell wall expansion often appears to be anti-correlated with the main orientation of cellulose microfibrils, microtubule organization, and the consecutive cell wall reinforcement, is not (Jonsson *et al*, 2022, Daher *et al*, 2018). These observations support a phenomenological model of strain-based growth and stress-based stiffening for the cell wall.

We now introduce some of these ideas in the introduction, in the description of the model and, more importantly, in the discussion:

Lines 259-275: "While strain stiffening, a mechanism involving a non-linear response of cell walls to deformation, has been shown to regulate cell growth in the shoot apical meristem, our analytic and numerical approaches show that a coupling between strain-based growth and stress-based wall stiffening can produce an incoherent feedforward loop that can explain the dynamics of seed growth and size regulation. The dependence of our system to these two different modalities of mechanosensitivity is the cornerstone of our model. How stress and strain are perceived and transduced into biochemical signals remain open questions in plants. At the molecular scale, forces should be perceived through the deformation of mechanosensitive molecules. How molecular deformations undergone by specific wall components are related to mechanical fields (such as strain and stress) that the wall experiences at the macroscopic scale is another unsolved question. Addressing this will necessitate understanding the nature of the molecular sensors, and analyzing their integration into the wall as well as their structural properties. Nevertheless, both theoretical work and experimental observations suggest that cells can indeed perceive and process various types of mechanical signals, which could provide positional and directional cues during development. We thus predict that mechanosensitive motives similar to those we have characterized here in developing seeds could be ubiquitous regulators of plant organogenesis."

The authors then suggest that such a model could explain why WT seeds grow larger than the *iku2* mutant, which when measured with indentation, appears to have higher turgor pressure. In a stress

based model, lower pressure means lower stress, and a larger seed. Another explanation could be that the plant arrests growth early, and the increased pressure is just a response to that. There are no experiments to test if the pressure really is the determining factor there. I think such a claim requires a bit more evidence in that regard.

In the first version of the manuscript, the idea that it was the increase in endosperm pressure that was triggering the early stiffening of the testa and the restriction of growth in *iku2* seeds, and not the opposite, was supported by the fact *IKU2* is specifically expressed in the endosperm and that we could recover WT seed growth patterns in *iku2*-fitted simulations by decreasing the pressure of the endosperm. In this new version, we added more direct evidence that pressure is indeed the key factor controlling seed growth in the new Fig. 5. We developed a system to cultivate *Arabidopsis* fruits for up to 9 days so that we could alter the pressure within the seed by tuning the osmolarity of the medium with the non-metabolizable sugar sorbitol. We observed that when we increase the osmolarity of the medium, seed growth is reduced but lasts longer, so that the seeds end up larger (Fig. 5c). These observations are in agreement with our model and our interpretation of the *iku2* phenotype and further support the idea that endosperm pressure really is the determining factor in our system.

Specific comments:

Abstract: I don't think the term "incoherent feed forward loop" is correct in this context, it will probably confuse rather than assist most readers in understanding the model. I would recommend removing it as it does not really add anything.

In GRN, a type-1 IFFL occurs when gene A directly activates gene C but indirectly activates gene B which then represses gene C. This type of motif is known to confer interesting characteristics on the resulting systems, which can be used as pulse generators or as response accelerators depending on the parameters of activation of A and B. In our system, we have the same motif as seen in type-1 IFFL as pressure both promotes growth and indirectly inhibits it through testa stiffening. Furthermore, we also see that our system can, as for IFFL in GRN, generate a pulse-like output for seed growth depending on the parameters of the stiffening function. We thus used the term "mechanosensitive incoherent feed forward" because of these interesting parallels between our system and type-1 IFFL from GRN. The use of such concepts in mechanobiology is actually not new. It has already been shown that the control of anisotropic growth in plants also involves a mechanical feedback loop where anisotropic cell growth, which relies on the control of cellulose deposition by cortical microtubules, affects organ growth and shape, which can feedback on microtubule organization through mechanical forces (Hamant *et al*, 2008). As a more general perspective, we also think that using concepts borrowed from information processing and network analysis has a didactic value in explaining mechanobiology, and that a common framework could be developed to describe cell processing of information through biochemical and mechanical signals.

We now added some of these notions in the text:

Lines 90-97: In this model, the stress and strain borne by the testa are respectively proportional to pr and pr/k . The fact that the strain is inversely proportional to the effective stiffness k is the cornerstone of our model: an increase in stiffness will decrease strain and consequently curb growth. By promoting stiffening, pressure-induced stresses therefore have antagonistic effects on testa growth. We thus formalized this system, schematized in Fig. 1d, as an incoherent mechanosensitive feedforward mechanism because its motif is similar to that displayed by a type-1 incoherent feedforward loop (IFFL) in gene regulatory networks, and because WT seed growth pattern also resembles the pulse-like output produced by type1-IFFL (Supplementary Fig. 1c).

Lines 35: Considering the limitations of the work in reference (10), I am not sure that can be generally accepted as fact.

According to this comment, we moved this reference to the discussion and modified our text to the following:

Lines 277-279: "This echoes recent work performed in the shoot apical meristem suggesting that cell hydrostatic pressure may not be homogenous within the epidermis, and may affect growth more locally."

Lines 67: Why? What is the purpose of seeing if the process can be called an IFFL? Why does it matter?

Please see comment above

Line 70: "assimilated to a", perhaps "approximated as"

This has been corrected.

Line 75: The model should be better described here. The reader should not have to go to the supplement, a high level description of the model needs to be added here. At the minimum it should mention what the growth and the stiffening depend on. In the end the model is very simple, it stops growing at a certain stress, and stress directly depends on size and pressure.

Based on this comment, we have expanded the description and assumptions of the model in the main text with a more explicit description of the fundamental equations:

Lines 71-96:

"To this end, we approximated the testa to a linearly elastic spherical shell of radius R , constant thickness h and homogeneous effective rigidity K , (Fig. 1c). We considered two mechanosensitive mechanisms taking place within this system (i.e. the idealized testa): (i) stress-based cell wall stiffening and (ii) strain-based cell wall expansion. Taken together both mechanisms can be formalized as a set of two dimensionless coupled differential equations, eq(1); where the first line depicts a strain-based growth process while the second line depicts the stress-based stiffening process:

$$\begin{cases} \dot{r} = \left(\frac{pr}{k} - 1\right)_+ r \\ \dot{k} = \gamma(1 - k + \alpha h_\eta(pr, \rho)) \end{cases} \quad (1)$$

In eq.(1), $\left(\frac{pr}{k} - 1\right)_+$ and $h_\eta(pr, \rho)$ correspond to the two mechanosensitive functions we considered. The former depicts a threshold linear function while the latter corresponds to a classic Hill function, as commonly used in biochemical signaling pathways. The state of our system is described by a set of two dimensionless variables: $r=R/h$, a dimensionless measure of the radius of our system and k a dimensionless version of its effective rigidity k . Besides these two variables, eq(1) features four dimensionless parameters $\{\gamma, \alpha, \rho, \eta\}$ that quantify cell wall stiffening properties (speed, strength, threshold and steepness) relative to the growth process (see rationale in Supplementary model S1.4). Finally, the adimensional pressure P plays the role of an external control variable as it is not a property of the system itself (i.e. the testa) but an input in the system (See Supplementary model S1 for a detailed derivation of this set of equations).

In this model, the stress and strain borne by the testa are respectively proportional to pr and pr/k . The fact that the strain is inversely proportional to the effective stiffness k is the cornerstone of our model: an increase in stiffness will decrease strain and consequently curb growth. By promoting stiffening, pressure-induced stresses therefore have antagonistic effects on testa growth. We thus formalized this system, schematized in Fig. 1d, as an incoherent mechanosensitive feedforward mechanism because its motif is similar to that displayed by a type-1 incoherent feedforward loop (IFFL) in gene regulatory networks, and because WT seed growth pattern also resembles the pulse-like output produced by type 1- IFFL²¹ (Supplementary Fig. 1c)."

Our model is indeed very simple, but displays subtle dynamics. In reality, there is no "stress threshold above" which growth stops. Looking at the growth equation – first line of system (1) – one can see that there is a strain threshold below which growth stops. Growth only stops if stiffness increases faster (in a highly non-linear manner) than stress, such that strain, which integrates stress and stiffness, drops below the growth threshold. Finally, note that even if the stiffening function is non-linear, it still takes several days for the testa to stiffen, as we can see both experimentally and in simulations (Fig. 3). So the reduction of seed growth rate over time is progressive (as observed in Supplementary Fig. 1c and 5a), and does not stop abruptly as the ideal of a "maximal stress threshold" would suggest.

Lines 78: The authors need to explain why the growth stops (if always true). Is that necessarily true for all parameters? For example, it is true that for parameters found in ii) where the growth doesn't stop, that lowering the pressure cannot change the outcome?

As previously mentioned we have now expanded the description of our model in the main text and better explain why the system stops growing. There are two properties that can influence the putative convergence of the system towards a steady state: (i) the initial state (radius and stiffness) it starts from and (ii) the values of its parameters $\{\alpha, \gamma, \rho, \eta\}$. For some values of these parameters, no steady state exists at all, no matter the initial state considered. This is captured in Supplementary Figures 3b to g where we looked at the existence of steady states when only one parameter was changing and the others were fixed (at values corresponding to the simulation best-fitting WT data). The pink zones on panels e, f, g depict parameter sets allowing steady states to exist. But even in these cases, where steady states can theoretically be reached, not all initial states evolve towards such stable solutions. This fact is now illustrated by Fig. 1e where the flow chart in the state space depicts all of the possible dynamics, given a fixed parameter set (allowing steady state), but starting from various initial conditions. The take home message here is that even when steady states are achievable, initial states converging towards them correspond to a very limited portion of the state space. For all the simulations we performed thereafter, we therefore chose initial conditions in accordance with the geometry observed in our biological data and parsimonious assumptions concerning the testa effective stiffness, i.e. initially, the stiffness corresponds to its stationary non-mechanically-enhanced value; this is mentioned in Supplementary model description 3.

Lines 79-80: non-sequitur, I don't see how this follows. If it said reached a steady state at the same size, then this would make sense.

This is related to the previous point. Note that for a given set of parameters allowing steady states, these steady states are not unique but spread over a region of the state space. This region corresponds to the intersection (highlighted in yellow) of the dash blue sigmoid curve (characteristic of stiffening arrest) with the red upper triangle (where growth stops) on the new panel Fig. 1e.

Lines 81-90: If you make the cell stiffer based on stress, and stress is proportional to pressure and size, it makes sense that this should follow. Again, this should be explained.

Lines 81-90 initially reported the parameter space exploration we performed and the results it yielded. Again, although simple in concept, the behavior of our model is subtle and sometimes deceives intuition. The point here is that it is not sufficient to make cells stiffer based on stress (proportional to size) to regulate growth. Since this process is dynamic, time plays a central role: if stiffening is not fast enough or starts too late, the system will not be able to balance these antagonist mechanisms and will not converge. Conversely, if stiffening is too fast or happens too early, the system will not expand sufficiently. A posteriori, these conclusions could appear as rather obvious; but they emerge from a simple model with no a priori. We explored a parameter space where the time-related dimension (the τ -line) and the synchronization-related dimension (the β -line) are spread over 6 orders of magnitude. The sharpness of the peak along the β -line (third panel of Fig. 1f) is a striking demonstration that synchronization between growth and stiffening is the most selective property of the model and this was far from obvious when we started.

We made this point more explicit in the main text by adding the following sentence:

Lines 122-126: *“More importantly, the parameter space analysis also revealed that the ratio between growth and stiffening threshold (β/τ) was the most constrained parameter. The sharp peak at the value of 1 (right hand-side panel of Fig. 1f) demonstrates that stiffening must be late compared to growth but also that a tight synchronization between both processes is required for seed growth control through a mechanosensitive regulation.”*

Lines 92-94: This is a bit problematic. A strongly non-linear response based on stress that happens late means the model is programmed to stop when it hits a certain stress. Since stress is directly proportional to size, the model really just stops growing when it hits a certain size. This would require that the cell have a method of sensing stress (independent of strain) which is not physically possible. There is no method to measure stress that does not involve strain.

Depending on the initial conditions the system starts from, it will not reach the same steady state. The final size is therefore not “programmed” within these equations, it emerges from them. Concerning the discussion about strain/stress measurement, we addressed this point in response to a previous comment and added a few lines about it in the discussion.

Lines 112-128: If growth essentially shuts down when a threshold stress is reached, then lower pressure would give larger seeds, as stress is an increasing function of both pressure and size. An alternate hypothesis is that growth is simply reduced in the mutant, and that pressure is increased because the plant tries to fight that.

See general comment above. Our new experiment involving the growth of seeds *in vitro* in media of different osmolarity supports our interpretation of the *iku2* mutant phenotype, in which seeds are smaller because pressure is higher, rather than the opposite interpretation.

Lines 137: Doesn't this argue that pressure is increased as a response to early stiffening?

ELA1 being mechanosensitive (i.e. as its expression can be induced by the application of mechanical forces), we expect that the increase of *ELA1* expression seen in *iku2* is a consequence of the enhanced pressure. We agree with the reviewer that testa stiffening could feedback on endosperm pressure, which would be very interesting. However, we do not have evidence for such a mechanism yet.

What happens in the *ela* mutants? I guess the plants are bigger so the seeds are probably bigger as well, but does that also mean the whole plant is size controlled by a similar mechanism? Leaves, stems, etc. Seems unlikely.

ELA1, which encodes an enzyme degrading the growth hormones gibberellins, is involved in the control of the growth of many plant organs as shown in a double mutant *ELA1-RNAi/ela2* by Zhang and colleagues (2011), but we do not know if *ELA1* can respond to mechanical forces in other organs than the seed. As a result, we do not know whether *ELA1* response to forces could control the growth of other plant organs. More generally, our model is based on two main assumptions:

1. That growth is promoted by the pressure of inner tissues and restricted by the mechanical properties of an outer layer.
2. That the restriction of growth by the outer layer depends on the perception of the tension induced by the pressure of the inner tissues on the outer layer.

The first assumption of our model could apply to other organs as it has been proposed that this type of control, which was termed “the epidermal growth control theory”, is applicable to various plant organs (Kutschera and Niklas, 2007). It is often the epidermis that restricts growth by stiffening its outermost cell wall (Hamant *et al*, 2008, Kierzkowzki *et al*, 2012). The second assumption would need to be tested for our model to apply to other organs. However, we know that mechanical forces can affect various cellular processes in the epidermis of many plant organs, including notably gene expression (See Landrein and Ingram, 2019 for a review on mechanical responses in different plant organs). We could thus imagine that epidermal cells could restrict growth in response to tension in other plant organs and that our model could thus be applicable in these contexts.

What about polyploids?

We have not studied polyploids but this could indeed be a very interesting perspective. Our model provides a framework to explain how cell mechanical properties and response to forces affect organ growth. We could thus test in the future if the mechanical properties (pressure or cell wall rheology) or the response of cells and tissues to forces is different in polyploids, thus explaining the increased size of the organ. In the case of the seed, we know that increasing the relative proportion of maternal or paternal genome in the endosperm has opposite effects on growth. With our model, we could hypothesize that these effects could be linked to changes in endosperm pressure (maybe as a result of changes in osmolyte accumulation) that then impact testa mechanical properties.

Lines 147: I would say that the evidence supporting the idea that PME_s promote growth is pretty weak.

We agree with the reviewer that there are many cases like ours where pectin demethylesterification is linked to a reduction of cell growth. However, we still wanted to be transparent about the fact that there are cases where the opposite has been proposed. Nevertheless, based on this comment we modified our text to the following:

Line 170-172: “In some cases, this process can promote enzymatic HG degradation, weakening the cell wall and promoting growth. However, fully demethylesterified HGs often form calcium-dependent cross-links that increase wall stiffness and inhibit growth.”

Lines 163-176: The authors state that they want to see if the testa is stiffer, and test this by determining the failure point during indentation. This does not probe stiffness, but rather strength of the layer.

We agree with this comment and we modified the text to explain the limitation of this technique and to state that we are not directly probing wall stiffness.

Like 185-188: "As wall 3 is an internal wall embedded within the testa, it is not possible to use an atomic force microscope to directly quantify the rigidity of the wall in vivo. However, we tested if the changes in wall composition we observed in iku2 correlated with changes in wall resistance to rupture by indentation."

They say they observe more force for iku2 seeds than WT at all indentation depths, although no force-indentation curves are shown. They then say that this could represent turgor pressure or stiffness differences, although the former would seem more likely, since they are doing rather large indentations. They do not say if they indent seeds of a similar size, so if the iku2 is both smaller and stiffer, that could indicate significantly higher turgor pressure, since a larger structure should appear stiffer at the same pressure. In any case these experiments don't really address what is suggested in the opening sentence of this section.

We have provided examples of force indentation curves in WT and iku2 seeds (without or with rupture of wall 3), that are now shown in Supplementary Fig. 11a. We did not see a strong correlation between the shape of the indentation curve and number of walls that were ruptured. As a result, the force that we show in Supplementary Fig. 11b is not the force measured when wall 3 is ruptured (as we do not know when this occurs), but corresponds to the force at the maximum indentation (30 μ m, 40 μ m or 50 μ m). We apologize if this was not very clear in our first manuscript. As the reviewer suggested, it is indeed likely that this force is higher in *iku2* because the pressure of the endosperm is higher (as we show in Fig.2c). It could also be linked to the fact that *iku2* seeds are smaller at 6 DPA (Fig.2c) so that the relative deformation we induce in *iku2* is higher. However, despite the fact that both indentation force and depth relative to organ size are higher in *iku2* compared to the WT, we see less rupture of wall 3, supporting even more strongly the idea that wall 3 of *iku2* is more resistant than in WT.

To better explain these points, we modified the text to the following:

Lines 191-198: "For all indentation depths, more force was needed to indent iku2 seeds than WT seeds (Supplementary Fig. 11b), which is likely due to the increased endosperm pressure we measured in iku2 at 5-6 DPA (Fig.2d). We also observed that the frequency of testa wall failure correlated with the indentation depth, and that walls 1 and 2 were more easily ruptured than wall 3 (Fig. 3i and Supplementary Fig. 11c). Finally, with an indentation of 40 μ m, we observed that the wall 3 of iku2 seeds was significantly less prone to rupture than that of the WT (p-value < 0.00001 in a chi2 test, Fig.3k), even though iku2 seeds are smaller and the force needed to perform a 40 μ m indentation in iku2 was almost 20% higher than that required in the WT."

The authors might consider to do osmotic treatments on the seeds. This could give an indication of the pressure difference. Indentation on plasmolyzed seeds could also indicate if bending stiffness of the testa is a factor for the larger observed indentation stiffness in *iku2*.

As shown in the article by Beauzamy and colleagues (2016), the rigidity of the seed is mostly due to the turgor pressure of the endosperm, so that plasmolyzed seeds are extremely soft. It would thus be very difficult to interpret the results of indenting such a structure.

Supplement:

Equation SE2: At this point it might be good to mention that this equation applies to the deformed configuration, that is after it is pressurised.

We agree with the reviewer and we have modified the text as follows:

"Given the spherical symmetry of our representation, this assumption yields the Laplace law, equation (SE2), relating tensile stresses, at mechanical equilibrium in the deformed configuration, within the seed coat, to the endosperm pressure and the geometrical properties of the shell"

Text after SE3: I don't think the Lockhart or Ortega models were developed for FEM analysis. Ref (4) Boudon et al. 2015 is almost the same model as Bassel et al. 2014, PNAS.

Our intent was not to say that the Lockhart or Ortega models were developed for FEM but that their strain-based derivations proposed in Boudon et al 2015 were. We apologize for the confusion. To resolve the ambiguity of our initial formulation, we propose the following amendment:

“This empirical law can be seen as an extension of the original Lockhart (1) and Ortega (2) models.”

We also thank the reviewer for pointing out the paper by Bassel and co-workers in PNAS 2014. Although they are indeed based on the same conceptual ideas, and both Bassel 2014 and Boudon 2015 make use of FEM to estimate numerically mechanical equilibrium of tissues, the growth equation used in Bassel 2014 (equation S24 p.3 of the SI document) is not totally equivalent to eq. (SE3) proposed here. However we agree that it deserves to be mentioned at this point, we therefore modified the text as follows:

“Such a strain-based update of the seminal Lockhart equation has already been used in previous modelling work (3-5) to account for experimental observations, namely that cells expand orthogonally to the cell wall's stiffest direction.”

N.B.: Ref. 3 points at Basel et al 2014

Cell wall Stiffening section: "de degradation term".

This typo has been corrected, we thank the reviewer for pointing it out.

Reviewer #1 (Remarks to the Author):

The revised manuscript by Creff and collaborator has been improved and partially addressed the issues that I have raised in my feedback. However, I still have main concerns about some important data shown in this paper:

1. Measurement of the turgor pressure is critical to support the model proposed in this paper. The model is based on the observation that turgor pressure decreases in WT and is stable and higher in iku2. We can see this effect clearly only in one replicate (main figure). The variability of the turgor pressure measurements between WT and iku2 replicates was explained by potential issues with the measurement method, changes in plant hydration, or in response to intrinsic or environmental signals. If the methodological issues, intrinsic or environmental signals can have so strong effects on differences in the measurements of turgor pressure, then what does it imply to the models. If authors would modify the input parameters to what they observe in the data, would they still have the same modeling output? If the variability between experiments is so strong, I would expect to provide more replicates to really see what happens with turgor pressure. In my opinion, the proposed model is not supported by currently provided data on the dynamics of turgor pressure unless additional replicates are provided to discriminate between different scenarios (i.e. stable or decreasing turgor pressure in iku2).
2. "Fig.3C actually displays the result of a single independent experiment. As the settings and growth conditions used were slightly different between the two independent experiments, we did not pool them even though the result was similar. Consequently, one experiment is shown in the main figures and one in supplementary data." – To ensure high research standards, a minimum of three independent replicates should be provided for each experiment. Especially in the case of experiments where "conditions used were slightly different". Also, what does it mean slightly different conditions and why were those experiments not performed in the same conditions?

Reviewer #2 (Remarks to the Author):

I'll address only my comment that the authors should try a mutant which is defective in seed coat stress-induced stiffening and examine the effect of the extra turgor of iku2 mutants in this background. Intuitively if we accept the authors' premise that increased stress induced stiffening limits seed size in iku2, and that ap2 mutants are defective in stress induced stiffening, then iku2 ap2 seeds should be similar in size to ap2 seeds or perhaps larger than ap2 seeds. However this is not what the authors find, they find that the two phenotypes are simply additive. I'm puzzled by this and not immediately clear that it supports the authors conclusions, despite the authors' statements in the text.

Instead I wonder if I should conclude from this that AP2-dependent testa stiffness and the IKU2 effect on seed size are instead through unrelated processes? I wonder if you have tried parametrising a model for ap2 seed size based on reductions in stress-induced stiffness parameters and then added in the iku2 turgor increase? If so does the model predict the observed double mutant phenotype? If so this would be quite convincing.

Reviewer #3 (Remarks to the Author):

I am generally satisfied with the changes that the authors have made to address the points raised in the review. It is nice to have added a more complete description of the model, and more experimental support for some of the hypothesis, as well as clarifications in several areas. I would still advise the authors against using the term "stress-based wall stiffening" (line 25?), as there is no physical realistic mechanism for that. If mechanical stiffening of the wall does occur, the obvious mechanisms are based on strain, for example fibres aligning as the wall is stretched. Since in an isotropic spherical shell strain directly follows stress, there is no difference in model outcome between strain stiffening and stress stiffening in this context, so why propose the less physically realistic possibility?

The authors write: "At the molecular scale, forces should be perceived through the deformation of

mechanosensitive molecules." - this statement is not really correct, deformations of all known mechanosensitive molecules are induced by strains, so trying suggest we don't know if it can be stress or strain is a bit misleading.

I still don't think the comparison with IFFLs in GRNs is helpful. IFFLs are a bit over-hyped in the GRN world at the moment, and I understand why the authors might want to exploit that hype, but I think it just muddies the presentation. The essence of their model lies in the mechanical feedback of stiffening of the wall on the growth of the wall, resulting in growth cessation. Feed forward control does not involve feedback, nor does it involve a loop. The "loop" of an IFFL comes from the incoherent part, that is the dual pathways with opposing regulation, but neither involve feedback. So although it looks like a loop on a network diagram, there is no loop in information flow, it only goes one way. This is why it is misleading to call your model an IFFL.

Although the authors point to other papers proposing models based on stress sensing, increasing the volume of papers is not going to make it any more plausible. In the end it is up to authors how they want to present their work. Ditto for the discussion around IFFLs.

Reviewer #1 (Remarks to the Author):

The revised manuscript by Creff and collaborators has been improved and partially addressed the issues that I have raised in my feedback. However, I still have main concerns about some important data shown in this paper:

1. Measurement of the turgor pressure is critical to support the model proposed in this paper. The model is based on the observation that turgor pressure decreases in WT and is stable and higher in *iku2*. We can see this effect clearly only in one replicate (main figure). The variability of the turgor pressure measurements between WT and *iku2* replicates was explained by potential issues with the measurement method, changes in plant hydration, or in response to intrinsic or environmental signals. If the methodological issues, intrinsic or environmental signals can have so strong effects on differences in the measurements of turgor pressure, then what does it imply to the models. If authors would modify the input parameters to what they observe in the data, would they still have the same modeling output? If the variability between experiments is so strong, I would expect to provide more replicates to really see what happens with turgor pressure. In my opinion, the proposed model is not supported by currently provided data on the dynamics of turgor pressure unless additional replicates are provided to discriminate between different scenarios (i.e. stable or decreasing turgor pressure in *iku2*).

According to the reviewer comments and as requested, we added a 4th independent experiment where we compared the pressure in Col-0 and in *iku2* seeds. As in previous experiments, this experiment showed that the pressure within *iku2* seeds is higher than that in WT seeds (Supplementary Fig. 5d). As requested we also tested what would happen if we do not consider *iku2* pressure as constant but as decreasing to a lesser extent than in the WT. To do so, we extracted time-dependent pressure functions for Col-0 and *iku2* from the experimental data (pooling the 4 independent experiments where we extracted endosperm pressure from stiffness measurements) and used these functions (together with intermediate ones) as inputs in simulations. The outcome of these simulations is the same as that observed when we simulated *iku2* with a constant pressure, namely that the size of the seed is proportional to the strength of the pressure drop. We moved the simulations of *iku2* that were done at constant pressure to the supplementary figures and replaced them in the main figures with the simulations performed using the pressure-drop functions extracted from the data. We also amended the text, according to the reviewer's comment, so that we do not state that *iku2* pressure is stable anymore but rather that it slightly decreases over time, even though these two scenarios give the same output in simulations.

2. "Fig.3C actually displays the result of a single independent experiment. As the settings and growth conditions used were slightly different between the two independent experiments, we did not pool them even though the result was similar. Consequently, one experiment is shown in the main figures and one in supplementary data." – To ensure high research standards, a minimum of three independent replicates should be provided for each experiment. Especially in the case of experiments where "conditions used were slightly different". Also, what does it mean slightly different conditions and why were those experiments not performed in the same conditions?

We have carried out this project over the course of 6 years and unfortunately, the growth chambers have been renovated during this time. As described in the material and methods, we initially grew plants under constant light so that we could minimize the effect of the diurnal cycle on plant growth. However, this growth condition was not available to us after the renovation of our growth chambers. As a result, we had to perform our experiments in long day cycles instead. We thus repeated, at least once, each experiment under long day conditions to ensure that there was no effect of the change of the light conditions on the outcome of our experiments. Regarding the experiment described in Fig.3C, which

corresponds to the measurements of *ELA1* expression using the transcriptional reporter *pELA1::VENUS*, we now added a new independent experiment that we had carried in a constant light chamber. It also supports the conclusion that *ELA1* expression is increased in *iku2* (although, this time, we could measure a statistical difference at heart-stage only). To further support that this result is robust in long days, we have also added the results of additional qPCR experiments (3 experiments performed on independent batches of plants containing 4 to 5 replicates each, all carried out under long day conditions) showing that *ELA1* expression is increased in *iku2* at 5 DPA in this growth condition (which was already shown for plants grown under constant light in Creff *et al*, 2015).

Reviewer #2 (Remarks to the Author):

I'll address only my comment that the authors should try a mutant which is defective in seed coat stress-induced stiffening and examine the effect of the extra turgor of *iku2* mutants in this background. Intuitively if we accept the authors' premise that increased stress induced stiffening limits seed size in *iku2*, and that *ap2* mutants are defective in stress induced stiffening, then *iku2 ap2* seeds should be similar in size to *ap2* seeds or perhaps larger than *ap2* seeds. However this is not what the authors find, they find that the two phenotypes are simply additive. I'm puzzled by this and not immediately clear that it supports the authors' conclusions, despite the authors' statements in the text.

Instead I wonder if I should conclude from this that AP2-dependent testa stiffness and the IKU2 effect on seed size are instead through unrelated processes? I wonder if you have tried parametrizing a model for *ap2* seed size based on reductions in stress-induced stiffness parameters and then added in the *iku2* turgor increase? If so does the model predict the observed double mutant phenotype? If so this would be quite convincing.

We agree with the reviewer that this result appears puzzling at first. For this reason, we performed the simulations proposed above. As now shown in Fig. 4g, we looked at the effect of modulations of the stiffening parameters in Col-0 and *iku2* simulations (using the pressure drop function mentioned above). These simulations showed that seed size in simulations is most sensitive to variations in the parameter ρ , which corresponds to the threshold between relative growth and stiffening. These simulations also show that increasing ρ by 10 to 20% in Col-0 drop function simulations increases the size of the seed in a similar manner to that observed in the *ap2* mutant; while similar increases in ρ in *iku2* drop function simulations allowed seeds to reach a size comparable to Col-0 seeds, as observed experimentally in the *iku2 ap2* double mutant. From a biological perspective, these simulations support the idea that the force-dependent stiffening of the testa is reduced but not abolished in *ap2*, which could explain why *ap2* seeds are actually not that large compared to WT seeds and why *ap2 iku2* mutant seeds are not larger than *ap2* seeds.

Reviewer #3 (Remarks to the Author):

I am generally satisfied with the changes that the authors have made to address the points raised in the review. It is nice to have added a more complete description of the model, and more experimental support for some of the hypothesis, as well as clarifications in several areas. I would still advise the authors against using the term "stress-based wall stiffening" (line 25?), as there is no physical realistic mechanism for that. If mechanical stiffening of the wall does occur, the obvious mechanisms are based on strain, for example fibres aligning as the wall is stretched. Since in an isotropic spherical shell strain directly follows stress, there is no difference in model outcome between strain stiffening and stress stiffening in this context, so why propose the less physically realistic possibility?

We think the term "stress-based stiffening" is relevant because the mathematical function we are considering takes the stress variable σ as input. However, we agree with the reviewer that

in a 1D model such as ours, defining the mechanosensitive stiffening process as a strain-based or stress-based function would have probably yielded similar results. However, we chose a stress-based formulation for two main reasons. First, the literature supports that the best-described mechanosensitive-stiffening pathway (*i.e.* microtubule-dependent cellulose deposition) is more likely to respond to stress than to strain. Secondly, in a multi-dimensional case where strain, stress and rigidity must be described as tensor fields, only a stress-based stiffening mechanism is able to generate anisotropic shapes. We amended the discussion according to the reviewer comment to clarify this point and mention these two reasons (second paragraph of the discussion).

The authors write: "At the molecular scale, forces should be perceived through the deformation of mechanosensitive molecules." - this statement is not really correct, deformations of all known mechanosensitive molecules are induced by strains, so trying suggest we don't know if it can be stress or strain is a bit misleading.

Indeed, in mechanics, forces and deformations are two sides of the same coin; one cannot exist without the other. Being geometric by nature, deformation appears more tangible and intuitive. This might explain why we tend to favour, even unconsciously, these variables when it comes to grasping concepts. Stress and strain are well defined and relevant in a large-scale perspective, encompassing the cell wall as a continuum. At this scale, molecules composing the wall and the cytoplasmic membrane are not considered individually and their behavior is averaged. Conversely, at the molecular scale, where the cell wall is described as a complex network of intertwined discrete elements (polymers and macromolecules), stress and strain fields cannot be defined and the proper notions to consider are forces, applied to macromolecules (in place of stress) and their deformations (in place of strain). So cell wall strain does not 'induce' deformation of molecules per se, there is no causality across scales. How can large-scale fields (strain, stress) be related to microscopic quantities (forces, deformations)? This question is a whole scientific field in itself and falls well beyond the scope of this manuscript. Such molecular mechanisms might be relying, in all likelihood, on complex biochemical and biophysical processes happening at the membrane/cell wall interface. Their mesoscopic outcomes are certainly numerous and subtle. However, at the mesoscopic scale, the preferred orientation of microtubules along the main stress directions and not the main strain direction, suggests that cells do have ways to "feel" large-scale stresses, regardless of the molecular apparatus involved.

I still don't think the comparison with IFFLs in GRNs is helpful. IFFLs are a bit over-hyped in the GRN world at the moment, and I understand why the authors might want to exploit that hype, but I think it just muddies the presentation. The essence of their model lies in the mechanical feedback of stiffening of the wall on the growth of the wall, resulting in growth cessation. Feed forward control does not involve feedback, nor does it involve a loop. The "loop" of an IFFL comes from the incoherent part, that is the dual pathways with opposing regulation, but neither involve feedback. So, although it looks like a loop on a network diagram, there is no loop in information flow, it only goes one way. This is why it is misleading to call your model an IFFL. Although the authors point to other papers proposing models based on stress sensing, increasing the volume of papers is not going to make it any more plausible. In the end, it is up to authors how they want to present their work. Ditto for the discussion around IFFLs.

We apologize if our point was not clear enough. In Fig. 1d we present a graph of interactions between variables of our model. Two triangular motifs can be identified in this graph:

- T_1 : stress (σ) \rightarrow strain (ϵ) \rightarrow radius (r) + radius (r) \rightarrow stress (σ).
- T_2 : stress (σ) \rightarrow strain (ϵ) + stress (σ) \rightarrow stiffness (k) \neg strain (ϵ).

Where \rightarrow means "induces" and \neg means "represses".

T_1 accounts for the growth mechanism and obviously represents a positive feedback loop. As the system grows, stress increases and promotes growth and so on... The T_2 motif, depicting the stiffening mechanism, also features two paths, both starting from stress and ending at strain, forming a closed loop. From a single stress input, the two paths mediate antagonistic effects on strain; they are thus labelled as incoherent. Independently of any "hype", this simplistic description of the interactions between the variables of our model matches the definition of an incoherent feedforward loop of type 1. In a feedforward loop there is, by definition, "no loop in information flow", for all arrows are pointing in the same direction, see Fig.1 of Sher-Orr (2002) or Figs. 1&2 of Mangan et al (2003) for instance.

The concept of regulatory motif (FBL, FFL...) is indeed not mandatory to analyze a regulatory network, be it genetic or mechanical. However, it summarizes a set of properties and enables one to grasp "the big picture" without focusing on the technical details. In this context, we think that it can bridge gaps between scientific communities working on different fields but sharing this common general regulatory network architecture. By using this terminology, our goal is precisely to promote this transversality and ease the understanding of plant morphodynamics for colleagues from related fields.

Reviewer #2 (Remarks to the Author):

I appreciate the authors' efforts in addressing the points I raised previously. I can see how most of the data are consistent with the proposed model, which may well be an accurate reflection of reality. But there are still some things that don't seem quite to add up. For instance why are iku2 seeds smaller than wt so early in development? Maybe I am struggling to understand how the model works but I don't quite grasp why early high pressure in wt doesn't cause the stress/strain issue, if indeed this is the cause of the small seed size in iku2 already observable a couple of days after pollination.

I think this is an interesting piece of work that will advance the field and stimulate debate. But I remain to be convinced it is the final word on the mechanism of seed size control by IKU2.

Reviewer #3 (Remarks to the Author):

Most of my comments from that last round still stand.

The arguments in the rebuttal in favor of stress sensing don't make sense to me. As I said previously, it is not physically possible to measure stress. Referring to other papers that support stress based mechanisms does not change the physics there. In this case the model would work fine assuming strain sensing, so it is a bit of an own-goal.

I also disagree with the discussion around IFFLs, and I think that the use of that genetic network terminology does not really help understanding in this context.

In the end it is up to the authors how they want to present their work.

Response to Reviewers

Reviewer #2 (Remarks to the Author):

I appreciate the authors' efforts in addressing the points I raised previously. I can see how most of the data are consistent with the proposed model, which may well be an accurate reflection of reality. But there are still some things that don't seem quite to add up. For instance why are *iku2* seeds smaller than wt so early in development? Maybe I am struggling to understand how the model works but I don't quite grasp why early high pressure in wt doesn't cause the stress/strain issue, if indeed this is the cause of the small seed size in *iku2* already observable a couple of days after pollination.

I think this is an interesting piece of work that will advance the field and stimulate debate. But I remain to be convinced it is the final word on the mechanism of seed size control by IKU2.

We thank the reviewer for his insightful comments. Regarding the two last comments above. The model supports the idea that the early high pressure in the WT does not induce the stress-dependent stiffening immediately after fertilization because the level of stress is still below the threshold of activation of the stress-stiffening pathway as shown in Fig.3a (because stress increase as a function of the radius of the seed, it also increases over time). Regarding *iku2*, the simulations predict that the stiffening pathway starts to be induced two days to three days after pollination (Fig.3a), which is consistent with the growth dynamics of *iku2* seeds (Fig. 2c). Although we did not look at the accumulation of demethylesterified at 2 DPA, we can already see a statistically significant difference in the intensity of the signal of the JIM5 antibody in wall 3 at 3DPA but not in the intensity of the signal of the other antibodies.

Reviewer #3 (Remarks to the Author):

Most of my comments from that last round still stand. The arguments in the rebuttal in favor of stress sensing don't make sense to me. As I said previously, it is not physically possible to measure stress. Referring to other papers that support stress based mechanisms does not change the physics there. In this case the model would work fine assuming strain sensing, so it is a bit of an own-goal.

We are sorry that the reviewer does not agree with our decision to favor a stress-sensing based model. However, we would like to highlight the fact that we did not merely refer to other papers that support stress-based mechanisms in our rebuttal, but that we also presented the argument that in a multi-dimensional case where strain, stress and rigidity must be described as tensor fields, only a stress-based stiffening mechanism is able to generate anisotropic shapes. We feel that this argument (to which the reviewer has not made any allusion) is, in itself, a valid basis for our decision.

I also disagree with the discussion around IFFLs, and I think that the use of that genetic network terminology does not really help understanding in this context.

Again, although we regret our difference of opinion with this reviewer, we feel that this (a difference of opinion) is what is being discussed here, rather than a fundamental error in the interpretation of our results. We would therefore like to maintain this analogy to IFFLs, which we, and other colleagues, have found useful in discussing our results.

In the end it is up to the authors how they want to present their work.